# Learning Stable Deep Dynamics Models for Partially Observed or Delayed Dynamical Systems

**Andreas Schlaginhaufen**[*]
ETH Zürich
andreas.schlaginhaufen
@epfl.ch

**Philippe Wenk**
ETH Zürich
wenkph@ethz.ch

**Andreas Krause**
ETH Zürich
krausea@ethz.ch

**Florian Dörfler**
ETH Zürich
dorfler@ethz.ch

## Abstract

Learning how complex dynamical systems evolve over time is a key challenge in system identification. For safety critical systems, it is often crucial that the learned model is guaranteed to converge to some equilibrium point. To this end, neural ODEs regularized with neural Lyapunov functions are a promising approach when states are fully observed. For practical applications however, *partial observations* are the norm. As we will demonstrate, initialization of unobserved augmented states can become a key problem for neural ODEs. To alleviate this issue, we propose to augment the system's state with its history. Inspired by state augmentation in discrete-time systems, we thus obtain *neural delay differential equations*. Based on classical time delay stability analysis, we then show how to ensure stability of the learned models, and theoretically analyze our approach. Our experiments demonstrate its applicability to stable system identification of partially observed systems and learning a stabilizing feedback policy in delayed feedback control.

## 1 Introduction

In this paper, we address the task of learning stable, partially observed, continuous-time dynamical systems from data. More specifically, given access to a data set $\{(t_0, y_0^l), \ldots, (t_N, y_N^l)\}_{l=1}^L$ of noisy, partial observations collected along $L$ trajectories of an unknown, stable dynamical system,

$$\begin{cases} \dot{z}(t) & = g\left(z(t)\right) & , z(t) \in \mathbb{R}^m \\ x(t) & = h(z(t)) & , x(t) \in \mathbb{R}^n, m \geq n, \\ y_i & = x(t_i) + \epsilon_i & , \epsilon_i \overset{\text{i.i.d.}}{\sim} \mathcal{N}(0, \sigma^2), \end{cases} \tag{1}$$

we would like to learn a model for the dynamics of $x(t)$. Moreover, it should be ensured that the model remains stable (we will be concerned with exponential convergence to 0) on unseen trajectories.

Learning such systems in a data-driven way is a key challenge in many disciplines, including robotics [Wensing et al., 2017], continuous-time optimal control [Esposito, 2009] or system biology [Brunton et al., 2016]. One powerful continuous-time approach to non-linear system identification are deep Neural ODEs (NODE), as presented by Chen et al. [2018]. Since neural networks are very expressive, they can be deployed in a variety of applications [Rackauckas et al., 2020]. However, because of that expressiveness, little is known about their system theoretical properties after training. Thus, there has been growing interest in regularizing such dynamics models to ensure favorable properties. In the context of ensuring stability of the learned dynamics, Kolter and Manek [2019] propose to jointly learn a dynamics model and a neural network Lyapunov function, that guarantees global stability via a projection method. Neural network Lyapunov functions have previously been employed by Richards et al. [2018] to estimate the safe region of a fixed feedback policy and by Chang et al. [2019]

---

[*]Correspondence to andreas.schlaginhaufen@epfl.ch, wenkph@ethz.ch.

35th Conference on Neural Information Processing Systems (NeurIPS 2021).

to learn a stabilizing feedback policy for given dynamics. Moreover, Boffi et al. [2020] prove that neural Lyapunov functions can also be learned efficiently from data collected along trajectories.

Thus far, all of these approaches are working directly with a standard ODE dynamics model. If the system's states are fully observed and the system is Markovian, this can be a valid choice. However, in many practical settings, partial observations and non-Markovian effects like hysteresis or delays are the norm. To address the limited expressivity of neural ODEs in the classification setting, Dupont et al. [2019] introduce Augmented Neural ODEs (ANODE). Here, a standard neural ODE is augmented with unobserved states, to extend the family of functions the model is able to capture. While Dupont et al. [2019] demonstrate that initializing the unobserved states at 0 is sufficient for the classification case, this is certainly not true when deploying ANODE as a dynamical system. In fact, our experiments in Section 4 demonstrate that learning this initial condition is a key problem in practice.

Inspired by state-augmentation methods in the time-discrete case, we thus propose to capture partial observability and non-Markovian effects via *Neural Delay Differential Equations (NDDE)*. NDDEs were very recently proposed in the context of classification by Zhu et al. [2021] and in the context of closure models for partial differential equations by Gupta and Lermusiaux [2020]. While NDDEs offer an elegant solution to avoid the Markovianity of neural ODEs, again little can be said about stability outside of the training set. In fact, our experiments in Section 4 show that in a sparse observation and high noise setting, a NDDE model that is stable on training trajectories may become unstable along new unseen trajectories. We therefore extend the ideas of neural network Lyapunov functions, originally developed for stability analysis of non-linear ODEs, to time-delay systems and introduce a Lyapunov-like regularization term to stabilize the NDDE. In contrast to ODEs, NDDEs have an infinite-dimensional state space, which requires careful discretization schemes we introduce in this work. We then showcase the applicability of the proposed framework for the stabilization of the NDDE model and for the task of learning a stabilizing feedback policy in delayed feedback control of known open loop dynamics.

In summary, we demonstrate the applicability of NDDEs to the case of modeling a partially observed dynamical system. We then leverage classical approaches for stability analysis in the context of delayed systems to develop a novel, Lyapunov-like regularization term to stabilize NDDEs. Furthermore, we provide theoretical guarantees and code for our implementation.[2]

## 2 Model and background

The main model of this paper, NDDEs, mathematically belongs to the class of *time-delay systems* that come with some additional difficulties compared to ODEs, both on the theoretical as well as the numerical side. Thus, we first recall some preliminaries and notation on time-delay systems. Then we continue with the model architecture and stability of time-delay systems.

### 2.1 Time-delay systems

Suppose $r > 0$ and consider the infinite-dimensional state space $\mathcal{C}_r := \mathcal{C}([-r, 0], \mathbb{R}^n)$ of continuous mappings from the interval $[-r, 0]$ to $\mathbb{R}^n$. Throughout this paper, we endow $\mathbb{R}^n$ with the Euclidean norm $\|\cdot\|_2$ and $\mathcal{C}_r$ with the supremum norm $\|\phi\|_r = \sup_{s \in [-r, 0]} \|\phi(s)\|_2$ for $\phi \in \mathcal{C}_r$. Further on, along a trajectory $x \in \mathcal{C}([-r, t_f], \mathbb{R}^n)$ we make use of the notation $x_t(\cdot) := x(t + \cdot) \in \mathcal{C}_r$ to denote the infinite-dimensional state at time $t \in [0, t_f]$. A subset $B \subseteq \mathcal{C}_r$ is referred to as *invariant* if $\gamma^+(B) = B$, where $\gamma^+(B) := \{x_t(\psi) \in \mathcal{C}_r : \psi \in B, t \geq 0\}$ denotes its positive orbit. For a locally Lipschitz function $f : \mathcal{C}_r \to \mathbb{R}^n$, an *autonomous time-delay system* is defined by the family of initial value problems:

$$\begin{cases} \dot{x}(t) &= f(x_t) \\ x(s) &= \psi(s), \ s \in [-r, 0], \ \psi \in \mathcal{C}_r. \end{cases} \tag{2}$$

In contrast to ODEs, the dynamics are given by a *Functional Differential Equation (FDE)* and the initial condition by a function $\psi \in \mathcal{C}_r$. As we are interested in autonomous dynamics, we will always set the initial time to zero. Furthermore, we denote by $x(\psi)(t) \in \mathbb{R}^n$ the solution and by $x_t(\psi) \in \mathcal{C}_r$ the history state at time $t$, starting from the initial history $\psi$. As discussed in Section 2.2, we will mainly focus on the important special case of *retarded delay differential equations with commensurate delays*

$$\dot{x}(t) = f(x(t), x(t - \tau), ..., x(t - K\tau)) = f(\mathbf{x}^\tau_{-K}(t)). \tag{3}$$

---

[2]Code is available at: https://github.com/andrschl/stable-ndde

Since some discretization is always necessary for computational tracktability, this implicitly includes a numerical approximation of general FDEs if the number of delays is chosen sufficiently large. It holds that $r = K\tau$, where for convenience we introduced the short-hand notation $\mathbf{x}^{\tau}_{-K}(t) := (x(t), x(t - \tau), \dots, x(t - K\tau))$. Note that while the instantaneous change (i.e., the vector field) in (3) depends only on a discrete set of observations $\mathbf{x}^{\tau}_{-K}(t)$, the initial history $\psi \in \mathcal{C}_r$ has to be given on the entire interval $[-r, 0]$ in order to have well-defined dynamics for $t \geq \tau$. As a consequence, in practice we need an interpolation of the initial history, and for numerical integration a specific DDE solver based on the method of steps [Alfredo Bellen, 2013] is required. Apart from this, existence and uniqueness of solutions to (2) and (3) follow in a similar fashion as for ODEs [Hale and Lunel, 1993, Diekmann, 1995].

## 2.2 Neural Delay Differential Equations

**Model architecture** The motivation of the model architecture is the following: We look for a general method to learn continuous non-Markovian time series which occur, for example, in partially observed dynamical systems. As already mentioned before, the temporal evolution of an ODE is uniquely determined by its current state, which makes NODEs inherently Markovian. Instead of augmenting NODEs with additional states, our approach is inspired by neural network based system identification of discrete-time dynamical systems: the latter copes with non-Markovian effects by augmenting the state space with past observations (i.e., literally memory states) in order to lift the problem back into a Markovian setting (see e.g. [Chen et al., 1990]). A continuous-time analog leads us to a FDE $\dot{x}(t) = f(x_t)$ where the current change is depending on the history $x_t(s) := x(t + s), \ s \in [-r, 0]$ up to some maximal delay $r > 0$. Since a neural network cannot represent a general non-linear functional $f$, we discretize the infinite-dimensional memory state $x_t$ as in Equation (3). This leads us to the NDDE model

$$\dot{x}(t) = f^{NN}_\theta (x(t), x(t - \tau), \dots, x(t - K\tau)) = f^{NN}_\theta \left(\mathbf{x}^{\tau}_{-K}(t)\right), \tag{4}$$

which is illustrated in Figure 1. Here, $f^{NN}_\theta$ is a feedforward neural network and $K$ the number of delays.

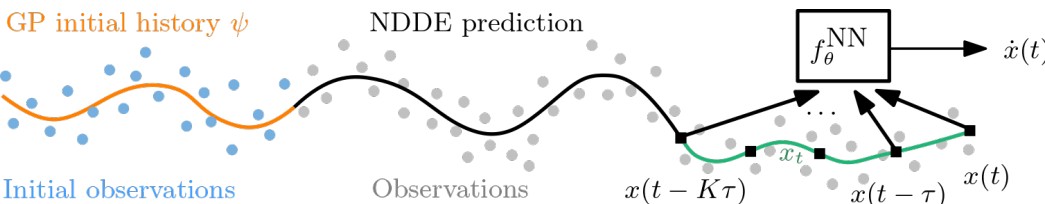

Figure 1: Graphical illustration of the NDDE model.

**Predictions** Given an initial history $\psi \in \mathcal{C}_r$ we integrate equation (4) to get the prediction at time $t$

$$\hat{x}(t) = \text{DDESolve}(\psi, f^{NN}_\theta, t_0, t). \tag{5}$$

However, in practice observations are subject to noise and cannot be sampled at an infinite rate. Hence, we need to approximate $\psi$ by a smoothed interpolation. For this purpose we employ Gaussian Process (GP) regression. Given a set $\{(t_0, y_0), \dots, (t_{N_{\text{hist}}}, y_{N_{\text{hist}}}\}$ of $N_{\text{hist}}$ observations along the initial history, we fit for each scalar initial history component a zero-mean GP. As a kernel, we choose the Radial Basis Function (RBF) kernel

$$k_{\gamma, \sigma_k}(t, t') = \sigma_k^2 \exp\left(-\frac{|t - t'|}{2l^2}\right) \tag{6}$$

with length-scale $l$ and kernel variance $\sigma_k^2$. This choice worked well in our experiments. Nevertheless, it is not crucial and other sufficiently smooth kernels such as Matérn 3/2 or Matérn 5/2 may be appropriate as well [Rasmussen and Williams, 2005]. For the smoothed interpolation of the initial history we are then using the posterior mean function,

$$\psi(t)_i = k_{tT} \left(K_{TT} + \sigma^2 I\right)^{-1} Y_i, \quad 1 \leq i \leq n, \tag{7}$$

where $Y_i = ((y_0)_i, \dots, (y_{N_{\text{hist}}})_i)$, $T = (t_0, \dots, t_{N_{\text{hist}}})$, $k_{tT} = (k(t, t_0), \dots, k(t, t_{N_{\text{hist}}}))$, and $K_{TT} = (k(t_j, t_k))_{j,k=1}^{N_{\text{hist}}}$. The kernel hyperparameters $l, \sigma_k^2$ as well as the observation noise variance $\sigma^2$ are estimated from data by marginal likelihood maximization.

**Training** For training we proceed similar to NODEs and minimize the least squares loss

$$J = \sum_{i=0}^{N} \|y_i - \hat{x}(t_i)\|_2^2 \tag{8}$$

along trajectories. While it is possible to utilize an interpolated continuous adjoint sensitivity method for calculating the loss gradients, differentiation through DDE solvers turned out to be significantly more efficient in our experiments. As discussed by Calver and Enright [2016], one reason is that jump discontinuities need to be accounted for that are later propagated in higher order derivatives along the solution of the adjoint state, and the DDE solver needs to be restarted accordingly. We therefore refrain from going into further details about adjoint methods and simply make use of the differentiable DDE solvers provided by Rackauckas and Nie [2017].

**Approximation capabilities** As opposed to neural ODEs, we are no longer learning an ordinary differential equation, but a retarded-type delay differential equation with constant delays. An interesting question is under which conditions a NDDE can model the time series corresponding to the partial observations $h(z(t))$ of the ODE system (1). As discussed in Appendix B, a sufficient condition for this is that the delay coordinate map,

$$E : \mathbb{R}^m \to \mathbb{R}^{(K+1)n},$$
$$z(t) \mapsto \mathbf{x}^{\mathcal{T}}_{-K}(t) = (h(z(t)), h(z(t - \tau)), \dots, h(z(t - K\tau))), \tag{9}$$

is one-to-one. For dynamical systems confined on periodic or chaotic attractors, the *delay embedding theorem* by Takens [1981] indeed shows that this holds true for large enough $K$ (for more details see Appendix B and the references therein). Although we do not assume that the system (1) is confined to such an attractor, our experimental results in Section 4 demonstrate approximation power and generalization capabilities of NDDEs when applied to dissipative systems.

## 2.3 Stability of time-delay systems

We discuss stability analysis for the general class of time delay-systems (2). We assume that the origin is an equilibrium, $f(0) = 0$, and slightly adjust the definition of exponential stability with respect to this equilibrium point as provided by Fridman [2014] to our needs:

**Definition 1** *For a fixed set of initial histories $\mathcal{S} \subseteq \mathcal{C}_r$ and constants $\gamma, M > 0$, we call system (2) $(\gamma, M)$-exponentially decaying on $\mathcal{S}$ over the time horizon $[0, t_f)$ if*

$$\|x(s)\|_2 \le M e^{-\gamma(s-t)} \|x_t(\psi)\|_r \quad \text{for } 0 \le t \le s < t_f, \forall \psi \in \mathcal{S}. \tag{10}$$

*For some invariant set $B \subseteq \mathcal{C}_r$ with $0 \in B$ the time delay system (2) is called $(\gamma, M)$-exponentially stable on $B$ if it is $(\gamma, M)$-exponentially decaying on $B$ over the time horizon $[0, \infty)$.*

Here, $\gamma$ measures the rate of decay and $M$ is an upper bound on the transient overshoot. In cases where we do not care about the specific values of $\gamma$ and $M$ we simply call the system (2) exponentially stable. Note that if a time-delay system is $(\gamma, M)$-exponentially decaying on a set of initial histories $\mathcal{S}$ over $[0, \infty)$, then it is also $(\gamma, M)$-exponentially decaying over $[0, \infty)$ on $\gamma^+(\mathcal{S})$. Since $\gamma^+(\mathcal{S})$ is invariant by definition, $(\gamma, M)$-exponential decay on $\mathcal{S}$ over $[0, \infty)$ is equivalent to $(\gamma, M)$-exponential stability on $\gamma^+(\mathcal{S})$.

**Razumikhin's method** A key method to prove exponential stability of non-linear ODEs are Lyapunov functions [Lyapunov, 1992]. However, directly applying ODE Lyapunov functions to time-delay systems leads to very restrictive results (e.g., a 1-dimensional system would not be allowed to oscillate [Fridman, 2014]). Nevertheless, along the same lines, two approaches geared towards stability analysis of non-linear time-delay systems exist. Whereas the method of Lyapunov-Krasovskii functionals [Krasovskii, 1963] is a natural extension of Lyapunov's direct method to an infinite-dimensional state space, the idea of Razumikhin-type theorems [Razumikhin, 1956] is to make use of positive-definite Lyapunov functions $V : \mathbb{R}^n \to \mathbb{R}_+$ with finite domains familiar from the ODE case and to relax the decay condition. Namely, a negative derivative of $V(x(t))$ at time $t$ is required only when we are about to leave the sublevel set $V^{\le \eta} = \{x \in \mathbb{R}^n : V(x) \le \eta\}$ of $\eta := \sup_{s \in [-r,0]} V(x(t + s))$. The following theorem establishes sufficient conditions for exponential stability.

**Theorem 1 ([Efimov and Aleksandrov, 2020])** *Assume there exists a differentiable function $V : \mathbb{R}^n \to \mathbb{R}_+$, positive reals $c_1, c_2, \alpha$, and a constant $q > 1$ such that along all trajectories starting in $\psi \in \mathcal{S} \subseteq \mathcal{C}_r$ the following conditions hold for all $t \in [0, t_f)$:*

*(i)* $c_1 \|x(t)\|_2^2 \leq V(x(t)) \leq c_2 \|x(t)\|_2^2$

*(ii)* $\dot{V}(x(t)) \leq -\alpha V(x(t))$ *whenever* $V(x(t+s)) \leq qV(x(t)) \quad \forall s \in [-r, 0]$.

*Then system* (2) *is* $(\gamma, M)$*-exponentially decaying on* $\mathcal{S}$ *over* $[0, T]$ *with decay rate* $\gamma = \min(\alpha, \frac{\log q}{r})/2$ *and* $M = c_2/c_1$. *Moreover, if for an invariant set* $B \subseteq \mathcal{C}_r$ *with* $0 \in B$ *conditions* $(i), (ii)$ *hold for all* $x_t \in B$ *then the time delay system* (2) *is exponentially stable on* $B$.

A function $V$ establishing stability of some invariant set $B$ by satisfying conditions $(i), (ii)$ in Theorem 1 is referred to as a Lyapunov-Razumikhin Function (LRF). However, due to the infinite dimension of $\mathcal{C}_r$ it is hard to verify the decay condition $(ii)$ on the entire state space $\mathcal{C}_r$. We thus focus on proving $(\gamma, M)$-exponential decay along trajectories starting within some fixed set of initial conditions $\mathcal{S} \subset \mathcal{C}_r$, which, as discussed before, is for an infinite time horizon equivalent to $(\gamma, M)$-exponential stability on $B = \gamma^+(\mathcal{S})$.

Note that Theorem 1 establishes sufficient, but not necessary conditions for exponential stability. The problem is that often it is not strong enough to check the Razumikhin condition

$$qV(x(t)) - V(x(t+s)) \geq 0 \tag{11}$$

only on the interval $s \in [-r, 0]$, but we should take into account more of the past observations of $V(x(t))$. It can therefore be helpful to reinterpret problem (2) as one in the state space $\mathcal{C}_{r_V}$ with some $r_V > r$ and to apply Theorem 1 to that problem. However, in this new – larger – state space, only initial histories of the form

$$\tilde{\psi}(s) = \begin{cases} \psi(s - (r - r_V)) & , s \in [-r_V, r - r_V] \\ x(\psi)(s - (r - r_V)) & , s \in [r - r_V, 0], \end{cases} \quad \text{with } \psi \in \mathcal{C}_r \tag{12}$$

need to be considered for stability in $\mathcal{C}_r$ [Hale and Lunel, 1993]. Furthermore, Proposition 1, which we prove in Appendix A, shows that also exponential stability follows, albeit at the price of a larger bound on the transient overshoot.

**Proposition 1** *If $f$ is $L_f$-Lipschitz, $f(0) = 0$, and $\psi, \tilde{\psi}$ defined as in* (12)*, then*

$$\left\| \tilde{\psi} \right\|_{r_V} \leq \|\psi\|_r \, e^{L_f(r_V - r)}. \tag{13}$$

Centred around this idea, necessary and sufficient Razumikhin-type conditions for discrete-time delay systems are given by Gielen et al. [2013]. In the following, we therefore treat $r_V$ as a hyperparameter that has to be chosen for the respective problem at hand.

## 3 Learning stable dynamics

We now propose an approach, based on neural LRFs, to enforce stability of a parametric DDE. The key idea is to jointly learn a neural network Lyapunov-Razumikhin function and the dynamics model. Similarly to [Richards et al., 2018, Chang et al., 2019] we propose to enforce stability via the loss function. This is in contrast to Kolter and Manek [2019] who use a projection-based approach to ensure stability in the forward pass. The main reason for this design choice is that a projective approach based on LRFs leads to discontinuities in the forward pass, which are problematic for DDE solvers [Alfredo Bellen, 2013]. Moreover, incorporating the Lyapunov neural network into the forward pass renders the model slow during inference time and a loss function based approach offers the opportunity to actively stabilize an initially unstable system, as we demonstrate in Section 4.

**Lyapunov neural network construction** Except for the decay condition along solutions (condition (ii) in Theorem 1), an LRF has the same form as an ODE Lyapunov function. We thus employ the same Lyapunov neural network as proposed by Kolter and Manek [2019]. The construction is based on an *Input-Convex Neural Network (ICNN)* [Amos et al., 2016]. The ICNN $x \mapsto g_\phi^{NN}(x)$ is convex by construction and any convex function can be approximated by such neural networks [Chen et al., 2019]. In order to satisfy the upper bound in condition $(i)$ of Theorem 1, the

activation functions $\sigma$ of $g_\phi^{NN}$ are required to additionally have slope no greater than one. To ensure strict convexity, and to make sure that the global minimum lies at $x = 0$, a final layer

$$V_\phi^{NN}(x) = \sigma(g_\phi^{NN}(x) - g_\phi^{NN}(0)) + c\,\|x\|_2^2 \tag{14}$$

is chosen. Here, $c > 0$ is a small constant. As for the activation function $\sigma$, having a global minimum at $x = 0$ requires $\sigma(0) = 0$. Furthermore, since we want to ensure Lipschitz continuity of the loss derivatives, we use a twice continuously differentiable smoothed ReLU version

$$\sigma(x) = \begin{cases} 0 & , x \leq 0 \\ \dfrac{x^3}{d^2} - \dfrac{x^4}{(2d)^3} & , 0 \leq x \leq d \\ x - \dfrac{d}{2} & , x > d. \end{cases} \tag{15}$$

This slightly differs from the original $\sigma$ proposed by Kolter and Manek [2019], since they only needed a once continuously differentiable one. This construction ensures that $V_\phi^{NN}(x) = \mathcal{O}(\|x\|_2^2)$ as $x \to 0$ and also $V_\phi^{NN}(x) = \mathcal{O}(\|x\|_2^2)$ as $\|x\|_2 \to \infty$. We can therefore always find constants $c_1, c_2$ such that the conditions (i) in Theorem 1 are satisfied. In the next step, we explain how to employ this neural network architecture to learn neural LRFs and at the same time stabilize a parametric delay differential equation of the form (3).

**Lyapunov-Razumikhin loss** As stated before, $V_\phi^{NN}$ satisfies condition $(i)$ in Theorem 1 by construction. The relaxed decay condition $(ii)$ however has to be enforced during training. Since it is practically infeasible to check the Razumikhin condition (11) on the continuous interval $[-r_V, 0]$, we need some discretization that still allows for stability guarantees. As we will analyze in this section, this is satisfied by the the following loss with discretized Razumikhin condition

$$\ell_{\mathrm{LRF}}\left(\phi, \theta, \mathbf{x}_{-K_V}^{\tau_V}(t)\right) =$$
$$\mathrm{ReLU}\left(\dot{V}_{\phi,\theta}^{NN}\left(\mathbf{x}_{-K_V}^{\tau_V}(t)\right) + \alpha V_\phi^{NN}(x(t))\right) \Theta\left(q V_\phi^{NN}(x(t)) - \max_{1 \leq j \leq K_V} V_\phi^{NN}\left(x(t - j\tau_V)\right)\right). \tag{16}$$

Here, $\Theta(\cdot)$ denotes the unit step function with $\Theta(s) = 1$ if $s \geq 0$ and $\Theta(s) = 0$ otherwise. Furthermore, for notational simplicity we choose $\tau_V \leq \tau$ and such that $\tau = l \cdot \tau_V$ for some integer $l \in \mathbb{N}$. According to Theorem 1, a zero loss $\ell_{\mathrm{LRF}}\left(\phi, \theta, \mathbf{x}_{-K_V}^{\tau_V}(t)\right)$ along a trajectory of length $[0, t_f)$ implies exponential decay along this trajectory. Moreover, if for a fixed set of initial histories $\mathcal{S}_{\mathrm{train}} \subseteq \mathcal{C}_r$ the loss (16) is zero along all trajectories starting in $\mathcal{S}_{\mathrm{train}}$ and over a time horizon $[0, \infty)$, then the delay differential equation is stable on $\gamma^+(\mathcal{S}_{\mathrm{train}})$. However, since we cannot check this for $t_f = \infty$, we choose $t_f$ large enough to ensure convergence to a sufficiently small region around the origin. Theorem 2 then also establishes exponential decay for trajectories starting not necessarily in – but close enough to – $\mathcal{S}_{\mathrm{train}}$. For its proof we refer to Appendix A.

**Theorem 2** *If the dynamics are $L_f$-Lipschitz and the LRF loss (16) is zero along trajectories starting in $\mathcal{S}_{train} \subset \mathcal{C}_r$ over a time horizon $[0, t_f)$, then the time-delay system is $(\gamma, M)$-exponentially decaying on $\mathcal{S}_{train}$ over $[0, t_f)$. Moreover, if for another set of initial histories $\mathcal{S} \supset \mathcal{S}_{train}$ and some $\varepsilon > 0$, the training set $\mathcal{S}_{train}$ is a $\delta$-covering of $\mathcal{S}$ (in the $\|\cdot\|_r$-norm) with $\delta = \varepsilon e^{-(L_f + \gamma)t_f}$, then the time delay system is $(\gamma, \bar{M})$-exponentially decaying on $\mathcal{S} \setminus B_\varepsilon(0)$ over the time horizon $[0, t_f)$ and with $\bar{M} = 2M + 1$. Here, $B_\varepsilon(0) = \{\psi \in \mathcal{C}_r : \|\psi\|_r \leq \varepsilon\}$ denotes the $\varepsilon$-ball around the origin.*

While for a zero loss exponential decay is guaranteed, the discretization of the Razumikhin condition might be introducing additional conservatism by requiring decay in $V_\phi^{NN}$ too often. However, Proposition 2 tells us that if the discretized Razumikhin condition holds, $\tau_V$ is small enough, and the current state lies outside of an $\varepsilon$-ball around the origin, then the continuous condition holds for some $\tilde{q} > q$. Furthermore, $\tilde{q}$ converges quadratically to $q$ as $\tau_V \to 0$. Remembering that the decay rate in Theorem 1 is $\gamma = \min(\alpha, \frac{\log q}{r})/2$, it becomes apparent that by discretization we are requiring a slightly larger rate of decay, which can however be controlled by the choice of $\tau_V$.

**Proposition 2** *Let $K_V$ and $\tau_V$ be such that $r_V := K_V \tau_V \geq r$ and let $x(\cdot)$ be a solution of $\dot{x}(t) = f(x_t)$ passing through $x_{t_0} \in \mathcal{C}_r$. Assume $\|x_{t_0}\|_{r_V + 2r} < \infty$ and $\|x_{t_0}\|_{r_V} > \varepsilon$. Furthermore, let $f$ be $L_f$-Lipschitz and differentiable. Then, if the discretized Razumikhin condition,*

$$V_\phi^{NN}(x(t_0 - k\tau_V)) \leq q V_\phi^{NN}(x(t_0)) \quad , \forall k \in \{0, 1, ..., K_V\}, \tag{17}$$

*is satisfied and $\tau_V$ is small enough, then the continuous Razumikhin condition,*

$$V_\phi^{NN}(x(t_0 + s)) \leq \tilde{q}(\tau_V) V_\phi^{NN}(x(t_0)), \tag{18}$$

*holds for any $s \in [-r_V, 0]$ and some $\tilde{q}(\tau_V)$ with $\tilde{q}(\tau_V) = q + \mathcal{O}(\tau_V^2)$ as $\tau_V \to 0$.*

The condition that the current state lies outside some $\varepsilon$-ball around the origin may be replaced by an assumption on the solutions' decay in $\|\cdot\|_{r_V}$. However, in practice we are usually satisfied with convergence to some small neighborhood of the origin, since, as already noted, we cannot choose an infinite time horizon and also as due to observation noise and data scarcity our model will always be subject to some modelling errors. We elaborate more on this issue and prove Proposition 2 in Appendix A.

**Stabilizing NDDEs** In order to stabilize an NDDE on a fixed set of initial conditions $\mathcal{S}_{\text{train}} \subseteq \mathcal{C}_r$ we minimize the stabilizing loss (16) on a set of data points $\{\mathbf{x}^{(1)}, \ldots, \mathbf{x}^{(N_{\text{LRF}})}\}$ with $\mathbf{x}^{(i)} \in \mathbb{R}^{n(K_V+1)}$ collected along trajectories starting in $\mathcal{S}_{\text{train}}$. The resulting gradients are added up with those from the NDDE loss (8). While this enables us to stabilize the NDDE on unseen trajectories, we still need an efficient method to generate realistic initial histories. Especially in the setting of partially observed systems we do not know much more about initial histories than that they are contained within a bounded, Lipschitz subset of $\mathcal{C}_r$. However, since for our NDDE model the initial history is given by a GP-mean function, it suffices to stabilize the NDDE for initial histories within the subset

$$\{\psi \in \mathcal{H}_k | \psi(t) = \sum_{i=1}^{N_{\text{hist}}} c_i k_{l,\sigma_k}(t, t_i)\} \tag{19}$$

of the reproducing kernel Hilbert space $\mathcal{H}_k$ corresponding to $k_{l,\sigma_k}(t, t')$. Furthermore, boundedness of initial histories translates into a bound on the norm of the expansion coefficients $\|(c_1, ..., c_{N_{\text{hist}}})\|_2 \leq A$ and Lipschitz continuity can be accounted for by upper bounding the inverse length-scale $1/l \leq B$ and the kernel variance $\sigma_k^2 \leq C$ [Rasmussen and Williams, 2005]. To satisfy these constraints, we sample at each training iteration initial histories $\psi \in \mathcal{S}_{\text{train}}$ as follows: The expansion coefficients $(c_1, ..., c_{N_{\text{hist}}})$ are sampled uniformly in an L2-ball, and $1/l, \sigma_k$ on bounded intervals $[0, B], [0, C]$, respectively.

Note, that while another possibility would be to integrate the loss (16) as a continuous regularization term into the NDDE loss, the discontinuities in (16) turn out to be problematic for DDE solvers.

**Delayed feedback control** The stabilizing loss (16) is essentially applicable to any parametric DDE of the form (3). Equations of this form also occur in delayed feedback control. Assume we want to learn a stabilizing state feedback $u(t) = \pi_\theta(x(t))$ for a known open loop control system $\dot{x}(t) = f(x(t), u(t - \tau))$ with input delay. In practice, such delays in the feedback loop are often introduced as a consequence of communication latencies and typically cause instability [Krstic, 2009]. The resulting closed loop system is a parametric DDE $\dot{x}(t) = f(x(t), \pi_\theta(x(t - \tau)))$, which can, for small enough delays, be stabilized in a data-driven way with our LRF loss (16). If the input delay exceeds some critical value, the system can no longer be stabilized by DDE methods and infinite-dimensional feedback taking into account the inputs history would be required [Krstic, 2009]. Experimental results for delayed feedback stabilization are provided in Section 4.

**Choice of hyperparameters** Our NDDE model (4) as well as the stabilizing loss (16) involve hyperparameters such as number and magnitude of delays, whose choice we discuss in the following. For our NDDE model, the number of delays $K$ clearly controls the representational capabilities. In general it is sufficient to choose $K$ large enough such that the delay coordinate map (9) is one-to-one. For periodic or chaotic attractors, Takens' Embedding Theorem 3 (see Appendix B) provides a sufficient lower bound on $K$ to ensure this. Moreover, in our experiments, larger values of $K$ ease training and – perhaps surprisingly – do not hurt generalization performance. Of course, an overly large number of delays leads to long training time per iteration, thus slowing down training again. Except for the first experiment where we directly compare NDDEs to ANODEs, we fix a relatively large number of delays $K = 10$ throughout the paper. While Takens' Theorem 3, is besides a periodicity condition, completely agnostic to the choice of the delay parameter $\tau$, various heuristics such as *Average Mutual Information* or *False Nearest Neighbours* exist in practice (for an overview see [Wallot and Mønster, 2018]).

With regard to the stabilizing loss (16), Proposition 2 proves that the number of delays $K_V$ controls the conservatism we introduce through discretization of the Razumikhin condition. Furthermore, as discussed in (12), the maximal considered delay $r_V$ controls the conservatism inherent to Razumikhin's Theorem 1 itself. Lastly, the parameters $\alpha$ and $q$ are directly related to the rate of decay $\gamma$ in Theorem 1 via $\gamma = \min(\alpha, \log q/r_V)$. We thus choose $\alpha \approx \log q/r_V$. Moreover, a too small choice of $K_V, r_V$ or too large choice of $\alpha, q$ can be detected via a non-zero LRF loss (16).

## 4 Experiments

**Learning partially observed dynamics** We first compare the applicability of Vanilla NDDEs and ANODEs for the task of learning a partially observed harmonic oscillator,

$$\dot{z}(t) = \frac{d}{dt}\begin{pmatrix} z_1(t) \\ z_2(t) \end{pmatrix} = \begin{pmatrix} 0 & 1 \\ -1 & 0 \end{pmatrix} z(t), \quad h(z(t)) = \begin{pmatrix} 1 & 0 \end{pmatrix} z(t). \tag{20}$$

We train the models over two training trajectories starting from $z_{0,1} = (1, 0)$ and $z_{0,1} = (0, 2)$ and with zero observation noise. For ANODEs we compare a model trained with given true augmented initial conditions (IC) against another model where we initialize the augmented states with zero and learn them via the adjoint method. Moreover, for the NDDE we compare a single delay model with $K = 1$ to a multiple delay model with $K = 10$. The resulting vector field plots for the ANODE models illustrated in Figures 2a-2c demonstrate that, whereas for true initial conditions the dynamics match the ground truth well, learning the augmented initial conditions turns out to be a key problem. In contrast, for our NDDE model the initialization is conveniently provided by the GP interpolation. This is also reflected in the learning curves in Figure 2d, where we see that both NDDE models yield a significantly lower train loss for fewer iterations compared to the ANODE models. Moreover, the NDDE with $K = 10$ achieves a better training score. For the rest of the experiments we therefore fix $K = 10$. For more information about the setup and additional experiments we refer to Appendix C.

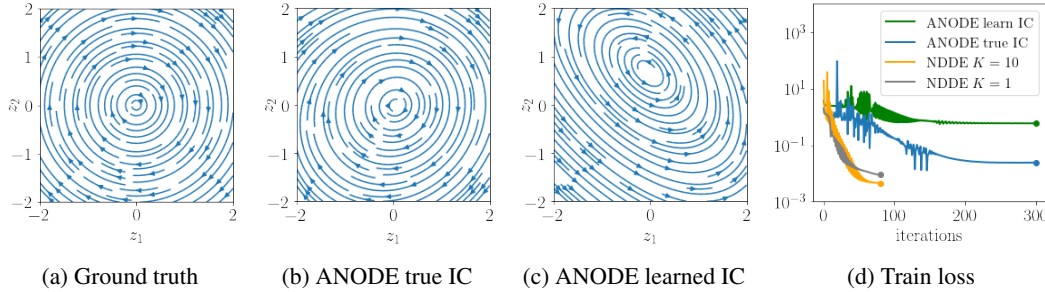

(a) Ground truth     (b) ANODE true IC     (c) ANODE learned IC     (d) Train loss

Figure 2: Comparison of ANODEs with true and learned initial conditions (IC) and Vanilla NDDEs. In (a)-(c), the phase portrait for ground truth and ANODE models are provided. Note, that since we are only interested in the first state the direction of rotation is irrelevant for the ANODE models. As it is impossible to draw a phase portrait of the NDDE model, the train losses for all four models are compared in Figure (d). While for given true initial conditions the ANODE model achieves a reasonable training fit, learning the augmented initial condition leads to a high training error. Moreover, both NDDE models show superior training performance compared to the ANODE models, both in terms of training error and number of iterations.

**Learning stable NDDEs** Kolter and Manek [2019] show that in the ODE case, a neural network dynamics model trained on stable data may become unstable for long-term prediction. For NDDEs, we observed this to be a problem in the setting of sparse observations, a high noise level, and generalization over initial histories. In particular, we consider a partially observed damped double compound pendulum, where only the angles of deflection $\varphi_1$ and $\varphi_2$, but not the angular velocities are observed. This is a complex non-linear dynamical system which, for low friction, exhibits chaotic behavior [Shinbrot et al., 1992]. The governing equations are derived in Appendix C.

For observation noise of variance $\sigma = 0.05$ and training and test data along 4 trajectories, we compare the generalization performance of a Vanilla NDDE and a NDDE stabilized with LRF regularization. We repeat the training for 20 independent weight initializations and noise realizations. The resulting predictions illustrated in Figures 3a-3b demonstrate that while the median prediction is stable, the upper 0.95 quantile explodes for the Vanilla NDDE. In contrast, the stabilized NDDE remains stable

on all test trajectories. Moreover, whereas the test loss in Figure 3c explodes for the unstable NDDE, the train losses are approximately the same. Thus, the LRF loss guides us to a stable optimum without sacrificing training performance.

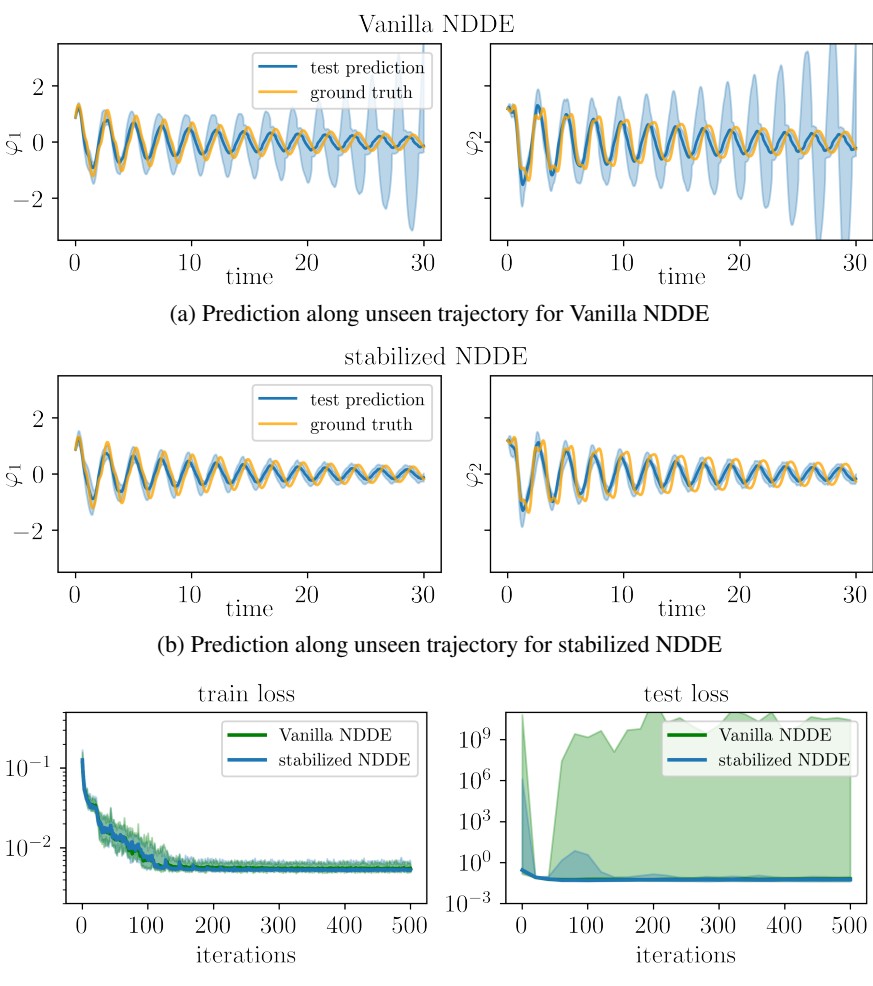

(a) Prediction along unseen trajectory for Vanilla NDDE

(b) Prediction along unseen trajectory for stabilized NDDE

(c) Train and test error of both models

Figure 3: In (a)-(b) the test predictions are shown for one of the test trajectories and in (c) train and test loss for all trajectories are illustrated. The lines indicate the median and the shaded area the 0.05 and 0.95 quantiles from 20 independent weight initializations and noise realizations.

**Stabilization with delayed feedback control** As a first application for learning a stabilizing feedback policy of a known open loop system, we consider a friction-less inverted pendulum with an input delay $\tau = 0.03$. The open loop dynamics are given by

$$\begin{pmatrix} \dot{x}_1(t) \\ \dot{x}_2(t) \end{pmatrix} = \begin{pmatrix} x_2(t) \\ \dfrac{g}{l}\sin(x_1(t)) + \dfrac{1}{ml^2}u(t-\tau) \end{pmatrix}. \tag{21}$$

Here, the states are $(x_1, x_2) = (\varphi, \dot{\varphi})$ where $\varphi(t)$ is the angle of deflection with respect to the fully upright position, $g$ indicates the acceleration of gravity, and $l$ and $m$ the length and mass of the pendulum. Furthermore, $u(t)$ is the torque which is applied at the pivot point. The goal is to learn a stabilizing feedback policy

$$u(t) = \pi(x(t)) = k_1 x_1(t) + k_2 x_2(t). \tag{22}$$

Similar to Chang et al. [2019], we initialize the parameters $k_1, k_2$ with the values from the Linear Quadratic Regulator (LQR) feedback policy calculated for the linearization of (21). For the training, we continuously generate new initial histories as follows: We sample ODE initial conditions on

a circle of radius $\pi/2$, assuming zero control for $t < 0$. Thus, the dynamics are described by an autonomous ODE along initial histories. As depicted in Figure 4a, the initially unstable state feedback can be stabilized by means of our Razumikhin loss. Furthermore, the speed of decay can be controlled by the choice of the hyperparameters $\alpha$ and $q$. Moreover, $\ell_{\text{LRF}}$ is zero along new test trajectories indicating that we indeed learned a valid LRF candidate for this set of initial histories.

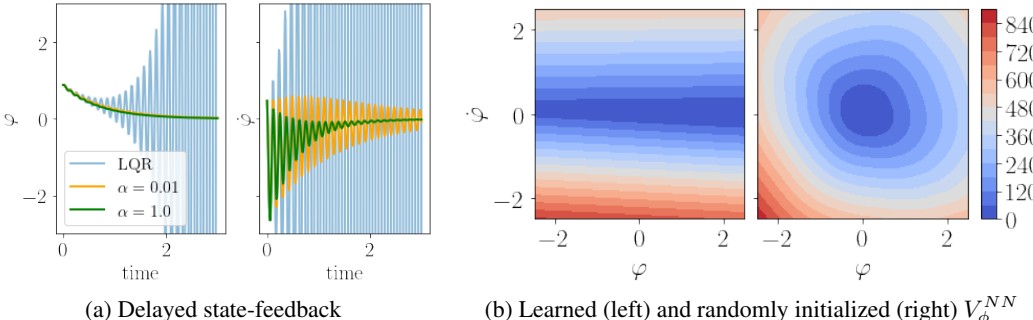

(a) Delayed state-feedback       (b) Learned (left) and randomly initialized (right) $V_\phi^{NN}$

Figure 4: In (a), the learned delayed state feedback policy is compared to the LQR control. The left plot in (b) shows the learned LRF candidate $V_\phi^{NN}$ and the right plot a random initialization. Whereas LQR is unstable for delayed feedback, the policies learned by minimization of the Lyapunov-Razumikhin loss are stable and the rate of decay can be controlled by the choice of the decay parameters $\alpha$ and $q$.

As a second – more complex – experiment, we consider stabilizing a cartpole with a delayed input force acting on the cart. In contrast to the two-dimensional inverted pendulum, this is a four-dimensional non-linear system. Its states are $x(t) = (\varphi(t), \dot{\varphi}(t), \xi(t), \dot{\xi}(t))$, where $\varphi$ again denotes the angle of deflection and $\xi$ the position of the cart. For the exact equations, we refer to [Stimac, 1999]. For the control force acting on the cart we assume a delay of $\tau = 0.05$ and aim at finding a stabilizing feedback policy $\pi_\theta(x(t))$. Similarly to the inverted pendulum experiment, Figure 5 shows that minimizing the LRF loss (16) enables us to find a stabilizing feedback policy from an initially unstable LQR feedback. Furthermore, the rate of decay can be controlled by the choice of the hyperparameter $\alpha$ and $q$.

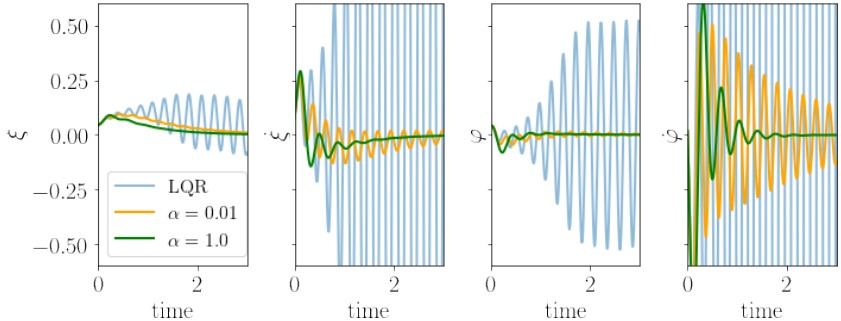

Figure 5: For the delayed cartpole, we compare the learned state feedback policy to the LQR controller. While LQR becomes unstable for delayed feedback, the feedback policies learned with the Lyapunov-Razumikhin loss are stable, and the decay rate can be controlled by the choice of $\alpha$ and $q$.

## 5 Conclusion

In this paper, we demonstrated that NDDEs are a powerful tool to learn non-Markovian dynamics occuring when observing a partially observed dynamical system. Via state augmentation with the past history we avoid the estimation of unobserved augmented states, which we showed to be a major problem of ANODEs when applied to partially observed systems. Based on classical time-delay stability theory, we then proposed a new regularization term based on a neural network Lyapunov-Razumikhin function to stabilize NDDEs. We further showed how this approach can be used to learn a stabilizing feedback policy for control systems with input delays. Besides experiments showcasing the applicability of our approach, we also provide code and a theoretical analysis.

## Acknowledgments

This research was supported by the Max Planck ETH Center for Learning Systems. This project has received funding from the European Research Council (ERC) under the European Union's Horizon 2020 research and innovation programme grant agreement No 815943 as well as from the Swiss National Science Foundation under NCCR Automation, grant agreement 51NF40 180545.

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
