# A Proofs

## A.1 Proof of Theorem 2

$(\gamma, M)$-exponential decay on the training set is a direct consequence of the LRF loss construction in (16) and Theorem 1. To show $(\gamma, 2M + 1)$-exponential decay on the set $\mathcal{S}$ we construct a coverage argument based on the following Lemma establishing continuous dependence of solutions:

**Lemma ([Smith, 2010])** *If the dynamics of the time-delay system (2) are $L_f$-Lipschitz it holds for all $\psi, \tilde{\psi} \in \mathcal{C}_r$:*

$$||x_t(\psi) - x_t(\tilde{\psi})||_r \leq e^{L_f t}||\psi - \tilde{\psi}||_r$$

For some $t \in [0, t_f]$ fix $\delta_t = e^{-(L_f + \gamma)t}\varepsilon \geq \delta$. Since $\mathcal{S}$ is a $\delta$-covering of $\mathcal{S}_{\text{train}}$ it is especially a $\delta_t$-covering. Therefore, for each initial history $\tilde{\psi} \in \mathcal{S}$ the training set contains an initial history $\psi \in \mathcal{S}_{\text{train}}$ with $||\tilde{\psi} - \psi||_r \leq \delta_t$. Thus,

$$||x(\tilde{\psi})(t)||_2 \overset{(i)}{\leq} ||x(\psi)(t)||_2 + ||x(\tilde{\psi})(t) - x(\psi)(t)||_2 \tag{23}$$

$$\overset{(ii)}{\leq} Me^{-\gamma t}||\psi||_r + ||x_t(\tilde{\psi}) - x_t(\psi)||_r \tag{24}$$

$$\overset{(iii)}{\leq} Me^{-\gamma t}(||\tilde{\psi}||_r + \delta_t) + ||x_t(\tilde{\psi}) - x_t(\psi)||_r \tag{25}$$

$$\overset{(iv)}{\leq} Me^{-\gamma t}(||\tilde{\psi}||_r + \delta_t) + e^{Lt} \tag{26}$$

$$\overset{(v)}{\leq} ||\tilde{\psi}||_r \left[ Me^{-\gamma t} + \frac{\delta_t}{||\tilde{\psi}||_r}(Me^{-\gamma t} + e^{Lt}) \right] \tag{27}$$

$$\overset{(vi)}{\leq} ||\tilde{\psi}||_r \left[ Me^{-\gamma t} + \frac{\delta_t}{\varepsilon}(Me^{-\gamma t} + e^{Lt}) \right] \tag{28}$$

$$\overset{(vii)}{\leq} ||\tilde{\psi}||_r \left[ e^{-\gamma t}M + e^{-(\gamma + L)t}(Me^{-\gamma t} + e^{Lt}) \right] \tag{29}$$

$$\overset{(viii)}{\leq} ||\tilde{\psi}||_r e^{-\gamma t}(2M + 1) = ||\tilde{\psi}||_r e^{-\gamma t}\tilde{M} \tag{30}$$

holds for all $t \in [0, t_f]$. Here, $(i)$ follows from the triangle inequality and $(ii)$ is a consequence of exponential decay on $\mathcal{S}$ and the definition of the $|| \cdot ||_r$-norm. In $(iii)$ we used that due to the reverse triangle inequality it holds $||\psi||_r \leq ||\tilde{\psi}||_r + ||\psi - \tilde{\psi}||_r \leq ||\tilde{\psi}||_r + \delta_t$. $(iv)$ follows by continuous dependence and in $(v)$ we rearranged terms. $(vi)$ holds since $\tilde{\psi} \notin B_\varepsilon(0)$ and $(vii)$ due to the definition of $\delta_t$. Finally $(viii)$ is a consequence of $e^{-x} \leq 1$ for $x \geq 0$.

## A.2 Proof of Proposition 1

For the proof we proceed similarly as Smith [2010] in their proof of continuous dependence. The main ingredient is the following form of the Grönwall-Bellman inequality:

**Lemma 1 ([Bellman, 1943])** *Given an interval $I = [a, b]$, two constants $A, B$ with $B \geq 0$, and a continuous function $u : I \to \mathbb{R}$. If*

$$u(t) \leq A + B \int_a^t u(s)ds, \ \forall t \in I, \tag{31}$$

*then it holds for all $t \in I$*

$$u(t) \leq Ae^{B(t-a)}. \tag{32}$$

Recalling that

$$\tilde{\psi}(s) = \begin{cases} \psi(s - (r - r_V)) & , s \in [-r_V, r - r_V] \\ x(\psi)(s - (r - r_V)) & , s \in [r - r_V, 0], \end{cases} \tag{33}$$

and $f(0) = 0$, we get for $t \in [r - r_V, 0]$,

$$\left\| \tilde{\psi}(t) \right\|_2 = \left\| \int_{r-r_V}^t f\left( x_{s-(r-r_V)}(\psi) \right) ds + \psi(0) \right\|_2 \tag{34}$$

$$\stackrel{(i)}{\leq} \int_{r-r_V}^t \left\| f\left( x_{s-(r-r_V)}(\psi) \right) ds \right\|_2 + \|\psi(0)\|_2 \tag{35}$$

$$\stackrel{(ii)}{\leq} \int_{r-r_V}^t L_f \left\| x_{s-(r-r_V)}(\psi) \right\|_r ds + \|\psi(0)\|_2 . \tag{36}$$

Here, we applied the triangle inequality in $(i)$ and $(ii)$ is a consequence of Lipschitz continuity. It therefore holds for all $t \in [r - r_V, 0]$,

$$\max_{\theta \in [-r_V, t]} \left\| \tilde{\psi}(\theta) \right\|_2 \stackrel{(i)}{\leq} \max_{\theta \in [r-r_V, t]} \int_{r-r_V}^{\theta} L_f \left\| x_{s-(r-r_V)}(\psi) \right\|_r ds + \|\psi\|_r \tag{37}$$

$$\stackrel{(ii)}{\leq} \int_{r-r_V}^t L_f \left\| x_{s-(r-r_V)}(\psi) \right\|_r ds + \|\psi\|_r \tag{38}$$

$$\stackrel{(iii)}{\leq} \int_{r-r_V}^t L_f \max_{\theta \in [-r_V, s]} \left\| \tilde{\psi}(\theta) \right\|_2 ds + \|\psi\|_r , \tag{39}$$

where $(i)$ is following from (36) and $\|\psi(0)\|_2 \leq \|\psi\|_r$, in $(ii)$ the term in the maximum is non-decreasing in $\theta$, and in $(iii)$ the maximum is taken over a larger interval than in $\|\cdot\|_r$. Defining $u(t) = \max_{\theta \in [-r_V, t]} \left\| \tilde{\psi}(\theta) \right\|_2$, the statement then follows from Lemma 1,

$$\left\| \tilde{\psi} \right\|_{r_V} = u(0) \leq \|\psi\|_r e^{L_f(r_V - r)}. \tag{40}$$

$\square$

## A.3 Proof Proposition 2

We start with bounding the deviation of $x(\cdot)$ from the linear interpolation between the observation points $\{(t_k := t_0 - k\tau_V, x_k := x(t_0 - k\tau_V))\}_{k=0}^{K_V}$. Lets denote the linear interpolation as $\tilde{x}(\cdot)$. Then by Rolle's Theorem [Phillips, 2003] we get the following standard upper bound on the norm of the interpolation error $e(t) := \tilde{x}(t) - x(t)$,

$$\|e(t)\|_2 \leq \frac{\tau^2}{8} \max_{s \in [t_0 - r_V, t_0]} \|\ddot{x}(s)\|_2 , \quad \forall t \in [t_0 - r_V, t_0]. \tag{41}$$

Using the Lipschitz continuity of $f$ and $f(0) = 0$, we get for the operator norm of the differential $\||Df|\| \leq L_f$ and $\|f(x_t)\|_2 \leq L_f \|x_t\|_r$. Furthermore, lets define $\rho := \|x_{t_0}\|_{r_V}$, some constant $C \geq \|x_{t_0}\|_{r_V + 2r}$, and $w \geq 1$ such that $\|x_{t_0}\|_{r_V + 2r} \leq w \|x_{t_0}\|_{r_V}$. Note, that since we assumed $\|x_{t_0}\|_{r_V} > \varepsilon$ we can always choose $w = C/\varepsilon$.

Then, applying the chain rule and using the above inequalities (41) simplifies to,

$$\|e(t)\|_2 = \leq \frac{\tau^2}{8} \max_{s \in [t_0 - r_V, t_0]} \left\| \frac{d}{dt} f(x_t)\big|_{t=s} \right\|_2 = \frac{\tau^2}{8} \max_{s \in [t_0 - r_V, t_0]} \||Df(x_s)|\| \cdot \|\dot{x}_s\|_r$$

$$\leq \frac{\tau^2}{8} L_f \max_{s \in [t_0 - r_V, t_0]} \max_{\xi \in [s-r, s]} \|f(x_\xi)\|_2 \leq \frac{\tau^2}{8} L_f \max_{s \in [t_0 - r_V - r, t_0]} L_f \|x_s\|_r$$

$$= \frac{L_f^2 \tau^2}{8} \max_{s \in [t_0 - r_V - 2r, t_0]} \|x(s)\|_2 \leq \frac{L_f^2 w \tau^2}{8} \max_{s \in [t_0 - r_V, t_0]} \|x(s)\|_2$$

$$= \frac{L_f^2 w \rho \tau^2}{8} , \tag{42}$$

for all $t \in [t_0 - r_V, t_0]$.

Now, we proceed with the derivation of (18) and assume that (17) holds. For convenience we define $V := V_\phi^{NN}$. Further on, we make use of the following claim, which we will prove later.

**Claim 1** : $\|\nabla V(x)\|_2 \leq M\rho$, $\forall x \in B_\rho(0, \|\cdot\|_2)$ with $M := (4c_2 - c_1)$

In particular, this means that $V$ is $(M\rho)$-Lipschitz in $B_\rho(0, \|\cdot\|_2)$. It then holds for any $s \in [-r_V, 0]$,

$$
\begin{aligned}
V(x(t_0 + s)) = V(\tilde{x}(t_0 + s) + e(t_0 + s)) &\overset{(i)}{\leq} V(\tilde{x}(t_0 + s)) + M\rho \|e(t_0 + s)\|_2 \\
&\overset{(ii)}{\leq} V(\beta x_k + (1 - \beta)x_{k+1}) + \frac{ML_f^2 w\rho^2\tau^2}{8} \\
&\overset{(iii)}{\leq} \beta V(x_k) + (1 - \beta)V(x_{k+1}) + \frac{ML_f^2 w\rho^2\tau^2}{8} \\
&\overset{(iv)}{\leq} qV(x(t_0)) + \frac{ML_f^2 w\rho^2\tau^2}{8}
\end{aligned}
$$

Here, in $(i)$ we used that $V$ is Lipschitz and $(ii)$ follows from (42) and the fact that $\tilde{x}(t + s)$ is a convex combination of two neighbouring data points. In $(iii)$ we used convexity of $V$ and $(iv)$ follows from the discretized Razumikhin condition (17).

To continue, let $x_m$ be such that $\|x_m\|_2 = \max_{s\in[t_0-r_V, t_0]} \|x(s)\|_2$ and $x^*$ such that $V(x^*) = \max_{s\in[t_0-r_V, t_0]} V(x(s))$. It then holds $V(x(t_0 + s) \leq V(x^*)$ and,

$$
V(x^*) \leq qV(x(t_0)) + \frac{ML_f^2 wV(x_m)\tau^2}{8c_1} \leq qV(x(t_0)) + \frac{ML_f^2 wV(x^*)\tau^2}{8c_1}.
$$

Therefore, if $8c_1 > ML_f^2 w\tau^2$, then for all $s \in [t_0 - r, t_0]$,

$$
V(x(t_0 + s)) \leq V(x^*) \leq \frac{q}{1 - ML_f^2 w\tau^2/(8c_1)} V(x(t_0)).
$$

Noting that $1/(1 - a\xi^2) = 1 + a\xi^2 + \mathcal{O}(\xi^4)$ as $\xi \to 0$ it follows that,

$$
V(x(t_0 + s)) \leq \tilde{q}(\tau)(V(x(t_0)) = q + \mathcal{O}(\tau^2).
$$

**Proof of Claim 1:** It only remains to proof the claim. For this purpose consider $x \in B_\rho(0, \|\cdot\|_2)$ and $h \in \mathbb{R}^n$ with $\|h\|_2 = 1$. We then have,

$$
c_2 \|x + \rho h\|_2^2 \overset{(i)}{\geq} V(x + \rho h) \overset{(ii)}{\geq} V(x) + \rho\nabla V(x)^\top h \overset{(iii)}{\geq} c_1 \|x\|_2^2 + \rho\nabla V(x)^\top h.
$$

Here, in $(i)$ and $(iii)$ we used the definition of $V$ and $(ii)$ follows from convexity of $V$. Rearranging terms and using the Cauchy–Schwarz inequality we arrive at,

$$
\begin{aligned}
\nabla V(x)^\top h &\leq \frac{1}{\rho}\left(c_2 \|x + \rho h\|_2^2 - c_1 \|x\|_2^2\right) \\
&= \frac{1}{\rho}\left((c_2 - c_1)\|x\|_2^2 + c_2\rho^2\|h\|_2^2 + 2\rho c_2 h^\top x\right) \\
&\leq (4c_2 - c_1)\rho,
\end{aligned}
$$

and since $h$ was an arbitrary element of the unit sphere it holds
$\|\nabla V(x)\|_2 \leq (4c_2 - c_1)\rho$. $\qquad\square$

The assumption $\|x_{t_0}\|_{r_V} > \varepsilon$ was needed to ensure that $\|x_{t_0}\|_{r_V+2r} \leq w \|x_{t_0}\|_{r_V}$ holds for some $w$. For exponentially decaying oscillations of the form

$$
x(t) = e^{-\gamma t}(a + b\cos(2\pi t/T_p)), \tag{43}
$$

there is no need for this assumption if we choose $r_V \geq T_p$, since

$$
\begin{aligned}
\|x_t\|_{r_V+2r} &\leq e^{-\gamma(t-r_V-2r)}(|a| + |b|) \\
e^{-\gamma t}(|a| + |b|) &\leq \|x_t\|_{r_V} \\
\Rightarrow \|x_t\|_{r_V+2r} &\leq w \|x_t\|_{r_V} \\
&\text{with } w = e^{\gamma(2r+r_V)}.
\end{aligned}
$$

Moreover, the choice $r_V \geq T_p$ is anyways a good idea, as it ensures that a local maximum of $V$ is contained in the interval where we check the Razumikhin condition.

## B  Delay embeddings

Assume we are given a dynamical system with $\mathcal{C}^2$ solution map,

$$\varphi_s : \mathbb{R}^m \to \mathbb{R}^m, \ z(t) \mapsto \varphi_s(z(t)) = z(t+s), \tag{44}$$

that is defined by a differential equation $\dot{z}(t) = g(z(t))$.

Furthermore, assume that $\mathcal{M} \subset \mathbb{R}^m$ is some submanifold that is invariant under $\varphi_s$ and let,

$$h : \mathbb{R}^m \to \mathbb{R}, z(t) \mapsto x(t) := h(z(t)),$$

be some $\mathcal{C}^2$ observation map. Now, we are interested in the question whether we can retain information about the state $z(t)$ from time-series measurements of $x(t)$. The delay embedding theorem by Takens [1981] provides us with conditions under which this can be answered positive. In particular lets define the delay coordinate map,

$$\begin{aligned}
E &: \mathcal{M} \to \mathbb{R}^d, \\
z(t) &\mapsto \mathbf{x}^\tau_{-d+1}(t) = (x(t), \quad x(t-\tau), \quad \dots, \quad x(t-(d-1)\tau)) \\
&= \left( h(z(t)), h \circ \varphi_{-\tau}(z(t)), \dots, h \circ \varphi^{d-1}_{-\tau}(x(t)) \right),
\end{aligned} \tag{45}$$

with sampling time $\tau$. Then the following theorem holds.

**Theorem 3 ([Takens, 1981])** *Let $\mathcal{M}$ be a compact manifold of dimension $M$ and suppose we have a dynamical system defined by (44) that is confined on this manifold. Let $d > 2M$ and suppose the periodic points of $\varphi_{-\tau}$ are finite in number, and $\varphi_{-\tau}$ has distinct eigenvalues on any such periodic point. Then the observation maps $h$, for which the delay coordinate map (45) is an embedding, form an open and dense subset of $\mathcal{C}^2(\mathcal{M}, \mathbb{R})$.*

Loosely speaking the above theorem tells us that if we consider enough delays in (45) and choose $\tau$ such that we do not hit too many periodic points, then for most observation maps $h$ the delay coordinate map $E$ is one-to-one on $\mathcal{M}$ and thus the inverse $E^{-1}$ exists on $E(\mathcal{M})$.

If $E^{-1}$ exists we have,

$$\begin{aligned}
\dot{x}(t) &= \frac{d}{dt} h(z(t)) = h'(z(t))\dot{z}(t) = h'(z(t))g(z(t)) \\
&= h'(E^{-1}(\mathbf{x}^\tau_{-d+1}(t)))g(E^{-1}(\mathbf{x}^\tau_{-d+1}(t))) = f(\mathbf{x}^\tau_{-d+1}(t)),
\end{aligned}$$

which is a DDE in $x(t)$. Due to the universal approximation property of neural networks [Cybenko, 1989] and provided that $x(t)$ is given on the interval $[t-(d-1)\tau, t]$, we can therefore represent $\{x(t)\}_{t \geq 0}$ by a NDDE.

Replacing $M$ with the upper box-counting dimension Theorem 3 can be extended to chaotic attractors [Sauer et al., 1991] and infinite-dimensional systems [Robinson, 2005].

## C  Experiments

### C.1  Remarks on implementation

During the experiments we use, for both the ANODE and the NDDE model, a fully connected depth six neural network architecture with hidden layer sizes $(32, 64, 128, 64, 32)$ for $f^{NN}_\theta$. Furthermore, the input and output layer sizes are chosen to match the respective model. As activation function we choose to use the Swish activation [Ramachandran et al., 2017] in favour of the standard hyperbolic tangent (tanh) activation function. Swish is a smoothed ReLU version, which consistently outperformed tanh in our experiments. For $V^{NN}_\phi$ we use an ICNN as described in (14) with hidden layer sizes $(64, 64)$.

Our code is based on the Julia libraries [Rackauckas and Nie, 2017] and [Rackauckas et al., 2020]. Moreover, we use a Tsitouras 5/4 Runge-Kutta method [Tsitouras, 2011] as ODE solver and a method of steps algorithm [Alfredo Bellen, 2013] based on the same ODE solver for the integration of DDEs. The experiments were run on a cluster using Intel Xeon Gold 6140 CPUs, none of them took longer than 2h.

## C.2 Supplementary experimental information

**Partially observed harmonical oscillator** For the comparison of ANODEs and NDDEs we trained on two training trajectories starting in $z_{0,1} = (1,0)$ and $z_{0,2} = (0,2)$. Moreover, we trained over a time horizon of $(t_0, t_N) = (0, 30)$ and used for each training trajectory a data set of $N = 150$ observations. Furthermore, we compare an NDDE model with $K = 10$ and $\tau = 0.3$ to a single delay model with $K = 1$ and $\tau = 2$. The training statistics are summarized in Table 1. We use exponentially decaying learning rates. The training predictions for all four models are illustrated in Figure 6.

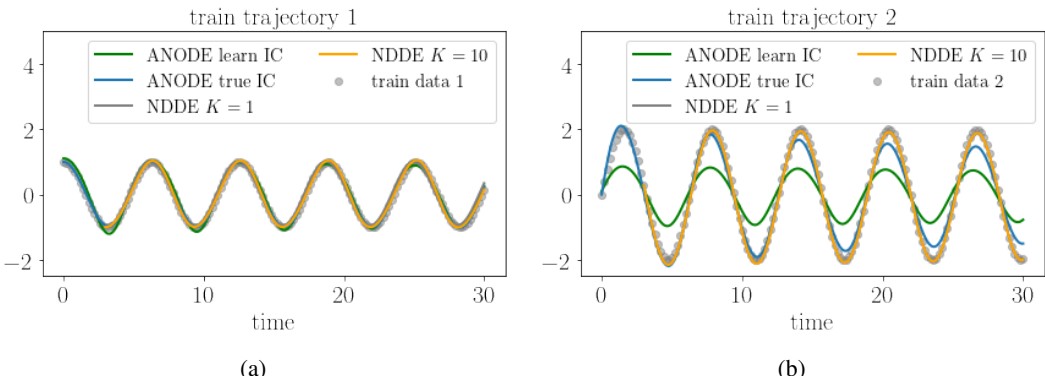

Figure 6: (a) shows the predictions along the first and (b) the predictions along the second training trajectory.

Table 1: Training summary harmonical oscillator

| model | wall time | iterations | learning rates | train MSE |
|---|---|---|---|---|
| ANODE true IC | 946.74 sec | 300 | 5e-3 - 1e-5 | 2.42e-2 |
| ANODE learned IC | 753.12 sec | 300 | 5e-3 - 1e-5 | 5.90e-1 |
| NDDE $K = 1$ | 151.75 sec | 80 | 5e-3 - 1e-5 | 8.99e-3 |
| NDDE $K = 10$ | 178.82 sec | 80 | 5e-3 - 1e-5 | 4.58e-3 |

**Learning stable 2-pendulum** We closely follow [Agarana and Akinlabi, 2018] to derive the equations of motion with Lagrangian mechanics. Position and squared velocity of the center of mass of the two connected rods are given by

$$x_1 = \frac{l}{2}\sin\varphi_1, \quad y_1 = -\frac{l}{2}\cos\varphi_1, \quad x_2 = 2x_1 + \frac{l}{2}\sin\varphi_2, \quad y_2 = 2y_1 - \frac{l}{2}\cos\varphi_1 \tag{46}$$

$$v_1^2 = \dot{x}_1^2 + \dot{y}_1^2, \quad v_2^2 = \dot{x}_1^2 + \dot{y}_1^2. \tag{47}$$

Accordingly, the potential $V$ and the kinetic energy $T$ are given by,

$$V = \sum_{i=1}^{2} m_i g y_i, \quad T = \frac{1}{2}\sum_{i=1}^{2}(m_i v_i^2 + I_i \dot{\varphi}_i^2). \tag{48}$$

Here, $m_i$ is the mass and $I_i = m_i l^2 / 12$ the moment of inertia with respect to the center of mass of rod $i$. Defining the Lagrangian $\mathcal{L} = T - V$ and the Rayleigh Dissipation Function

$$D = \frac{1}{2}\sum_{i=1}^{2} b_i \dot{\varphi}_i^2, \tag{49}$$

the corresponding Euler-Lagrange equations are

$$\frac{d}{dt}\left(\frac{d\mathcal{L}}{d\dot{\varphi}_i}\right) = \frac{d\mathcal{L}}{d\varphi_i} - \frac{dD}{d\dot{\varphi}_i}, \quad i \in \{1, 2\}. \tag{50}$$

We then use the symbolic algebra solvers provided by [Gowda et al., 2021, Ma et al., 2021] to solve for $\ddot{\varphi}_1, \ddot{\varphi}_2$. The resulting ODE is four dimensional, however we assume to only observe the positions $\varphi_1, \varphi_2$. Furthermore, we use the pendulum parameters $m_i = 1, l = 1, b_i = 0.1$ for $i \in \{1, 2\}$.

For the experiment, the Vanilla and the stabilized NDDE model are both trained on 4 training trajectories over 500 episodes. We use a cyclic learning schedule with repeated exponential decays between 5e-3 - 1e-6 and of period 50. The average training time for the stabilized NDDE model was 52min as opposed to 35min for the Vanilla NDDE. In each training step of the stabilized NDDE training, new initial histories are sampled and the stabilizing loss (16) is minimized along the corresponding trajectories by means of stochastic gradient descent. The training predictions in Figure 7 illustrate that the Lyapunov regularization is not significantly affecting the training fit. Moreover, both models proof to be robust to noisy observation in the training set. The training and model parameters for the NDDE training are summarized in Table 2 and for the stabilizing training in Table 3. Here, $T_i, N_i, L_i$ for $i \in \{$train, test$\}$ indicate the time horizon, the number of observations per trajectory, and the number of trajectories.

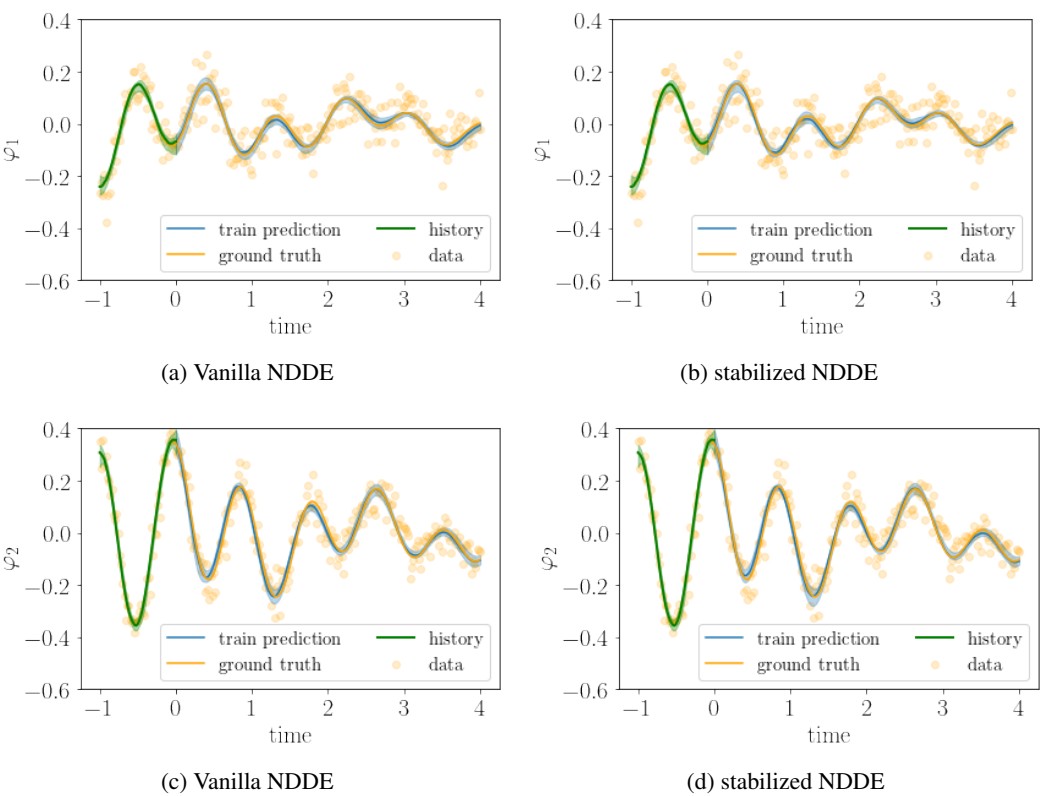

(a) Vanilla NDDE          (b) stabilized NDDE

(c) Vanilla NDDE          (d) stabilized NDDE

Figure 7: (a) and (c) show the train predictions along one of the 4 train trajectories for the Vanilla NDDE. (b) and (d) show the predictions for the stabilized NDDE. The shaded areas indicate 0.05 and 0.95 quantiles. Moreover, in each plot a single noise realization is depicted.

Table 2: NDDE training setup stable 2-pendulum

| $\tau$ | $K$ | $T_{\text{train}}$ | $N_{\text{train}}$ | $L_{\text{train}}$ | $T_{\text{test}}$ | $N_{\text{test}}$ | $L_{\text{test}}$ | batch time | batch size |
|---|---|---|---|---|---|---|---|---|---|
| 0.1 | 10 | 4.0 | 200 | 4 | 40.0 | 2000 | 4 | 200 | 4 |

Table 3: Stabilizing training setup stable 2-pendulum

| $\tau_V$ | $K_V$ | $T_{\text{Stab}}$ | $\alpha$ | $q$ | batch size |
|---|---|---|---|---|---|
| 0.1 | 20 | 10.0 | 0.01 | 1.01 | 256 |

**Inverted pendulum stabilization**  For the delayed feedback we choose a time delay parameter $\tau = 0.03$ and the parameters summarized in Table 4. Furthermore, we use exponentially decaying learning rates between 5e-2 - 1e-6 and minimize the LRF loss (16). In each episode we sample 4 new ODE initial conditions distributed on a circle of radius $\pi/2$ in order to get new ODE initial histories. The loss curves illustrated in Figure 8 demonstrate that the LRF loss (16) is indeed zero along new trajectories.

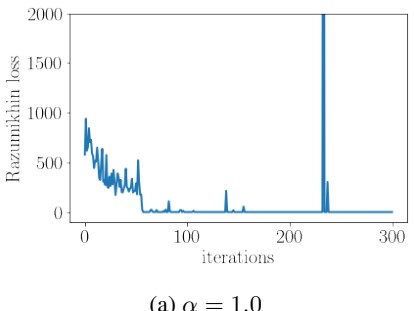

(a) $\alpha = 1.0$

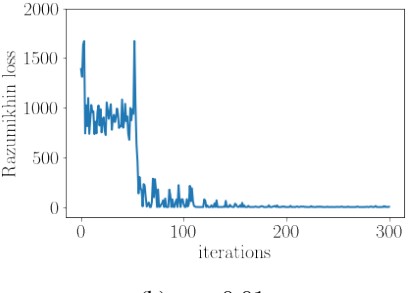

(b) $\alpha = 0.01$

Figure 8: Stabilizing loss along new trajectories.

Table 4: Stabilizing training setup inverted pendulum

| $\tau_V$ | $K_V$ | $T_{\text{Stab}}$ | batch size |
|---|---|---|---|
| 0.01 | 20 | 3.0 | 256 |

**Cartpole stabilization**  For the delayed feedback we assume a time delay $\tau = 0.05$ and the parameters summarized in Table 5. Furthermore, we use exponentially decaying learning rates between 5e-1 - 1e-5 and minimize the LRF loss (16). In each episode we sample 4 new ODE initial conditions distributed on a sphere of radius 0.1 in order to get new ODE initial histories. Furthermore, both feedback policies – trained with $\alpha = 0.01$ and $\alpha = 1.0$ – achieve a zero LRF loss on new trajectories at the end of training.

Table 5: Stabilizing training setup cartpole

| $\tau_V$ | $K_V$ | $T_{\text{Stab}}$ | batch size |
|---|---|---|---|
| 0.025 | 20 | 3.0 | 256 |

### C.3 Additional experiments

**Stable partially observed oscillator**    We consider a stable, partially observed harmonical oscillator defined by the differential equations

$$\dot{z}(t) = \frac{d}{dt}\begin{pmatrix} z_1(t) \\ z_2(t) \end{pmatrix} = \begin{pmatrix} 0 & 1 \\ -1 & -2\gamma \end{pmatrix} z(t), \quad h(z(t)) = \begin{pmatrix} 1 & 0 \end{pmatrix} z(t), \quad (51)$$

where we choose a damping coefficient $\gamma = 0.05$ and observation noise of standard deviation $\sigma = 0.3$. We are again comparing a Vanilla NDDE against a stabilized NDDE. Furthermore, we use the parameters summarized in Tables 6 and 7. Similarly to the 2-pendulum, the train predictions illustrated in Figure 9a and 9b match very closely. However, Figures 9c and Figures 9d are again showcasing that while the test predictions for the Vanilla NDDE explodes, the stabilized NDDE remains stable.

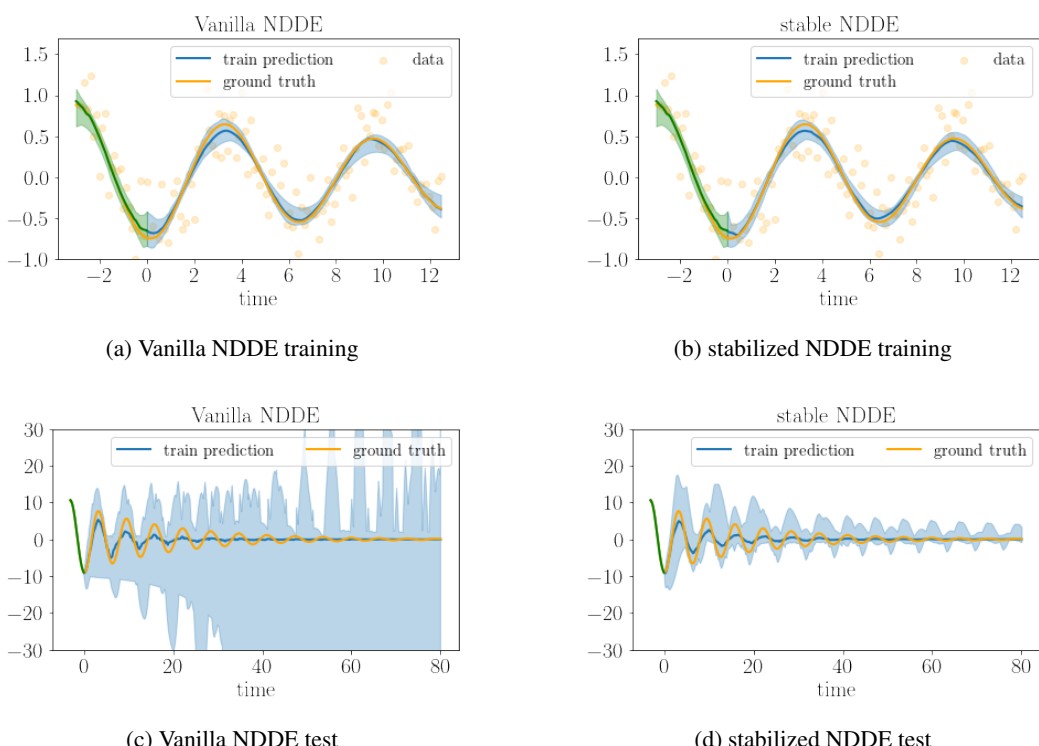

(a) Vanilla NDDE training                         (b) stabilized NDDE training

(c) Vanilla NDDE test                             (d) stabilized NDDE test

Figure 9: (a) and (b) illustrate the predictions along one of the training trajectories. The data points indicate one of the 20 noise realizations. In (c) and (d) the test predictions are plotted against the ground truth. The shaded areas are again indicating the 0.05 and 0.95 quantiles and the blue lines the median predictions.

Table 6: NDDE training setup stable oscillator

| $\tau$ | $K$ | $T_{\text{train}}$ | $N_{\text{train}}$ | $L_{\text{train}}$ | $T_{\text{test}}$ | $N_{\text{test}}$ | $L_{\text{test}}$ | batch time | batch size |
|---|---|---|---|---|---|---|---|---|---|
| 0.3 | 10 | $4\pi$ | 100 | 4 | $40\pi$ | 1000 | 4 | 100 | 4 |

Table 7: Stabilizing training setup stable oscillator

| $\tau_V$ | $K_V$ | $T_{\text{Stab}}$ | $\alpha$ | $q$ | batch size |
|---|---|---|---|---|---|
| 0.3 | 30 | 30.0 | 0.01 | 1.01 | 256 |

**Predator-prey dynamics**   As a last experiment we consider the the well-known Lotka–Volterra equations that model the population dynamics of a species of predators and its prey. The equations are given by,

$$\frac{dx}{dt} = \alpha x - \beta xy \tag{52}$$

$$\frac{dy}{dt} = -\gamma y + \delta xy. \tag{53}$$

Here, $x$ denotes the prey and $y$ the predator population. Moreover, the parameters $\alpha, \beta, \gamma, \delta$ describe the growth and death rates of the two species. We choose $\alpha = 5/3$, $\beta = 4/3$, $\gamma = \delta = 1$ and assume that we only observe the prey population $x(t)$. Further on, we use two training trajectories starting in $x_{0,1} = (2, 2)$ and $x_{0,2} = (3, 3)$ with 150 observations each and a time horizon of (0,20). For the NDDE we choose $\tau = 0.5$ and $K = 10$. In contrast to the former experiments we impose a hard 100min limit on the wall time and use mini-batching with a batch time of 50 observations and batch size 16 for the NDDE. Note, that for ANODEs batching is non-trivial when we strive to learn the initial conditions. The training predictions illustrated in Figure 10 and the MSEs in Table 8 again show superior performance of the NDDE in comparison with both the ANODE models. Moreover, similarly as for the harmonical oscillator, the ANODE with learned initial conditions performance worse than the model provided with true initial states.

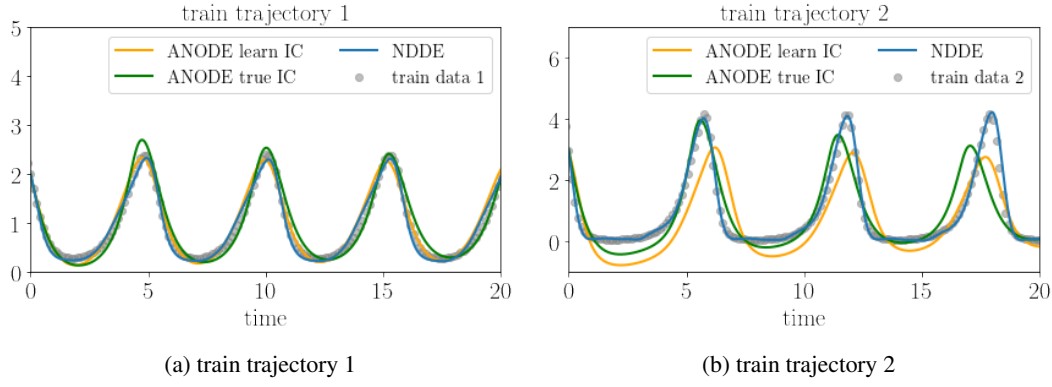

(a) train trajectory 1          (b) train trajectory 2

Figure 10: Predictions of the prey population $x(t)$ along the two training trajectories of the Lotka-Volterra system.

Table 8: Training summary Lotka-Volterra

| model | wall time | iterations | learning rates | train MSE |
|---|---|---|---|---|
| ANODE true IC | 100min | 437 | 5e-3 - 1e-5 | 0.305 |
| ANODE learned IC | 100min | 419 | 5e-3 - 1e-5 | 0.875 |
| NDDE | 100min | 998 | 5e-3 - 1e-5 | 0.023 |