# OpenReview forum: "Learning Stable Deep Dynamics Models for Partially Observed or Delayed Dynamical Systems"
_NeurIPS.cc/2021/Conference — NeurIPS 2021 Poster_

### Official Review · Reviewer_2gyi · 2021-07-14

**Rating:** 7
**Confidence:** 4

**Summary:**

This paper extends prior work on stable deep neural network dynamics models to learn stable models with partially observed states, and to learn (simple) control policies with time-delayed feedback. To achieve this, the authors minimize a Lyapunov-Razumikhin loss term when training _neural delay differential equations_. They demonstrate the capabilities of their system on a few tasks.


**Limitations And Societal Impact:**

The paper's technical limitations have been addressed in the review. This is basic research, and has no direct negative societal impact.

**Main Review:**

The work is the novel application of Lyapunov-Razumikhin stability to Neural DDEs. Applicable related work is cited.

## Quality

The derivation of the Lyapunov-Razumikhin loss term proceeds methodically from the background section. The special care taken with the Lyapunov NN construction and the LRF loss discretization is especially appreciated.

The authors are forthright in their evaluation of their own work: they clearly highlight potential pitfalls (such as how the stability certification is for each trajectory and not global, etc.)

### Experiments

Their choice of a two-link pendulum with partially-observed states is a reasonable evaluation task to highlight how the LRF loss prevents blow-up. This is only a single evaluation, though, and it is not obvious to me that ANODE (at least, when used with true initial conditions) will fail at solving the two-link pendulum.

Having more experiments would make this much better. In particular, if they can find a condition in which ANODE/vanilla NDDEs fail to learn but stable NDDEs succeed, this would further support their claim on LRF acting to regularize training (line 294). One way to do this would be to try this on three- or more-link pendulums as well.

The delayed feedback control experiment is well-selected.

The experiment tasks are all relatively simple and small. The paper would be much better served by presenting either a more complex “showcase” task, or a comparison to a common benchmark.

## Clarity

The submission is well-organized and mostly clear. Some key experimental details are in the appendix, but given that the focus of this paper is on the methodology, this is reasonable. (For the delayed feedback control experiment, I suggest mentioning the selection of $\tau=0.03$ in the experimental section itself.)

There is a little ambiguity in the creation of the initial history: in particular, lines 248-254, I’m not sure which data is used to train/fit the GP-mean, and if this is replaced/appended to at each iteration.

As this is a stability-by-loss-function approach, there is also some concern on how well the stability generalizes. (Answering the question “how likely are other trajectories produced by the model to be stable?”) Presenting some experiment quantifying this would alleviate this concern.

## Significance
This work joins a body of previous work on stable neural-network dynamics models using loss terms. Its ability to deal with partially-observable states and delayed control is somewhat interesting. The experimental results don’t showcase what the method could do in practice.

To make the results compelling depends on what this method can do in practice. Perhaps this could be showcased by performance on a more complicated common control-theory benchmark task (like a delayed or partially-observed cartpole?)

(Minor formatting issue: please consider allowing Latex to position figures instead of embedding them in text flow, It improves readability.)



**Time Spent Reviewing:**

5

---

> ### Author Response · Authors · 2021-08-10
> **Response to reviewer 2gyi**
>
> Thanks a lot for your positive feedback, valuable comments and helpful suggestions. In the following we would like to respond in detail to your comments. We addressed some of your concerns in our general comment to all reviewers, but would like to discuss some specific points you raised in the following.
>
> ## Our Comments
> - __(C)__ *"Blow up of Vanilla NDDEs is only shown on a single evaluation task.  Furthermore, it is not obvious to methat ANODE (at least, when used with true initial conditions) will fail at solving the two-link pendulum.Having more experiments would make this much better."* __(A)__ We  provide  a  second  blow  up  experiment in Appendix C.3, although for the less complex task of a stable, partially  observed harmonical oscillator. Concerning ANODEs: Since ANODEs with learned initial conditions achieved a poor training fit in our experiments (see Figure 2 and Figure 9), we decided to focus on NDDEs for the generalization experiments. However, in [1] blow up is also shown for a fully observed n-pendulum with neural network dynamics.
>
> - __(C)__ *"There is a little ambiguity in the creation of the initial history: in particular, lines 248-254, I’m not sure which data is used to train/fit the GP-mean, and if this is replaced/appended to at each iteration."* __(A)__ Thanks a lot for pointing this out. In our experiments the initial histories that are sampled for the regularization are resampled at each iteration. We will clarify this in line 254. These new initial histories sampled for the stabilizing loss are different from the ones learned from the training data. Is this sufficiently addressing your concern or have we missed something?
>
> - __(C)__ *"As this is a stability-by-loss-function approach, there is also some concern on how well the stability generalizes. (Answering the question “how likely are other trajectories produced by the model to be stable?”) Presenting some experiment quantifying this would alleviate this concern."* __(A)__ Yes, it is true that we only experimentally showcase the generalization performance of our approach. In Figures 3(d) we see that while the median test loss is not exploding for Vanilla NDDEs, blow up occurs for the 0.95 quantile. On the other side, all independently trained regularized NDDE models remain stable in this experiment. In order to theoretically treat generalization, we will add a new theorem after line 241 in the paper, which establishes exponential decay based on continuous dependence of solutions (as outlined in detail in the comment to all reviewers).
>
> - __(C)__ *"To make the results compelling depends on what this method can do in practice. Perhaps this could be showcased by performance on a more complicated common control-theory benchmark task (like a delayed or partially-observed cartpole?)"* __(A)__ Great idea, thanks! We have added a stabilization experiment for a delayed cartpole to the experiments section (see comment to all reviewers for more details).
>
> - __(C)__ *"(Minor formatting issue: please consider allowing Latex to position figures instead of embedding them in text flow, It improves readability.)"* __(A)__ Thanks for this hint. We will change this!
>
> - __(C)__ *"For the delayed feedback control experiment, I suggest mentioning the selection of $\tau = 0.03$ in the experimental section itself."* __(A)__ We changed this. Thanks for pointing this out!
>
> [1] Gaurav Manek, J. Zico Kolter, "Learning Stable Deep Dynamics Models"

---

### Official Review · Reviewer_Xxju · 2021-07-15

**Rating:** 6
**Confidence:** 3

**Summary:**

This paper aims to learn stable deep dynamics models for partially observed or delayed dynamical systems. The authors build their method upon Neural Delay Differential Equations (NDDEs) [1], where the main idea is to augment the state with the past history, thus avoiding the estimation of unobserved augmented states. To stabilize the learned dynamics, the authors then propose to combine NDDEs with a Lyapunov-like regularization term.

The authors show that NDDEs are powerful tools to model partially observed dynamical systems where the dynamics are non-Markovian. Furthermore, they demonstrate that the proposed method outperforms Augmented Neural ODEs (ANODE) in partially observed systems and is able to learn a stabilizing feedback policy for control systems with input delays.

[1] Qunxi Zhu, Yao Guo, Wei Lin, "Neural Delay Differential Equations"

**Limitations And Societal Impact:**

Please see my main review for the limitations, and I do not see any obvious potential negative societal impact of this work.

**Main Review:**

[Strength]

This paper targets an important research question of learning stable deep dynamics models for partially observable scenarios. Deep neural networks are powerful tools to fit the observation data, yet additional treatments have to be in place to ensure that the learned model satisfies certain properties like stability, energy preservation, etc.

This paper is on the line of adding structures to the learning procedure based on our understanding of the dynamical systems. The authors take a step forward by extending to partially observed or delayed systems, which I believe has the potential to apply to safety-critical systems where it is crucial to have guarantees on convergence.

The authors provide theoretical analysis and convergence guarantees on the proposed method.


[Weakness]

My primary concern of this paper is its novelty. This work builds on top of Neural Delay Differential Equations (NDDE) [1] by jointly learning a neural network Lyapunov-Razumikhin function. However, similar techniques for stabilizing dynamical systems with neural Lyapunov functions have already been investigated in [2, 3] and many others. Thus, to me, the proposed method seems to be a direct combination of existing methods, making this paper a bit insufficient in technical novelty.

My second concern is the lack of evaluation tasks. For evaluation, the authors only consider a simple hand-designed system, a partially observed damped double pendulum, and a friction-less inverted pendulum. However, for papers that line the similar direction like [3], they considered much more challenging cases, including wheeled vehicles and humanoid balancing. In [4], they investigated the n-link pendulum and high-dimensional systems like video texture generation. The current set of evaluating environments may be a bit too simple, limiting the significance and potential impact of this paper.


[1] Qunxi Zhu, Yao Guo, Wei Lin, "Neural Delay Differential Equations"

[2] Spencer M. Richards, Felix Berkenkamp, Andreas Krause, "The Lyapunov Neural Network: Adaptive Stability Certification for Safe Learning of Dynamical Systems"

[3] Ya-Chien Chang, Nima Roohi, Sicun Gao, "Neural Lyapunov Control"

[4] Gaurav Manek, J. Zico Kolter, "Learning Stable Deep Dynamics Models"


================================

Post-rebuttal comments

I have read the reviews from other reviewers and the rebuttal from the authors. I appreciate the authors' efforts in providing the additional experimental results and the detailed discussions about the novelty of the proposed method and the comparison with previous methods. I encourage the authors to include them in the revised version of the paper.

The rebuttal has sufficiently addressed my concerns, and I raised my score from 5 (Marginally below the acceptance threshold) to 6 (Marginally above the acceptance threshold).

**Time Spent Reviewing:**

4 hours

---

> ### Author Response · Authors · 2021-08-10
> **Response to reviewer Xxju**
>
> Thanks a lot for your feedback, valuable comments and helpful suggestions. In the following we would like to respond in detail to your comments. We addressed some of your concerns in our general comment to all reviewers, but would like to discuss some specific points you raised in the following.
> ## Our Comments
> - __(C)__ *"My primary concern of this paper is its novelty. This work builds on top of Neural Delay Differential Equations (NDDE) [1] by jointly learning a neural network Lyapunov-Razumikhin function. However, similar techniques for stabilizing dynamical systems with neural Lyapunov functions have already been investigated in [2, 3] and many others. Thus, to me, the proposed method seems to be a direct combination of existing methods, making this paper a bit insufficient in technical novelty."* __(A)__ We added a discussion about novelty into the comment to all reviewers. While the approaches you mention certainly inspire our work, there are several technical subtleties when porting to continuous time, which require significant innovations.
>
> - __(C)__ *"My second concern is the lack of evaluation tasks. For evaluation, the authors only consider a simple hand-designed system, a partially observed damped double pendulum, and a friction-less inverted pendulum. However, for papers that line the similar direction like [3], they considered much more challenging cases, including wheeled vehicles and humanoid balancing. In [4], they investigated the n-link pendulum and high-dimensional systems like video texture generation. The current set of evaluating environments may be a bit too simple, limiting the significance and potential impact of this paper."* __(A)__ It is correct that the authors of [3] and [4] are considering higher dimensional systems, although they investigate the much simpler fully observed and non-delayed setting. However, to address this point we added a stabilization experiment of a delayed cartpole (see comment to all reviewers for more details).
>
> [1] Qunxi Zhu, Yao Guo, Wei Lin, "Neural Delay Differential Equations"
>
> [2] Spencer M. Richards, Felix Berkenkamp, Andreas Krause, "The Lyapunov Neural Network: Adaptive Stability Certification for Safe Learning of Dynamical Systems"
>
> [3] Ya-Chien Chang, Nima Roohi, Sicun Gao, "Neural Lyapunov Control"
>
> [4] Gaurav Manek, J. Zico Kolter, "Learning Stable Deep Dynamics Models"

---

> > ### Author Response · Authors · 2021-08-23
> > **short follow-up**
> >
> > We would like to thank reviewer Xxju again for their helpful comments and feedback. Please let us know if the above response was sufficient and in case there are no further questions, we would appreciate if you could update your score.

---

### Official Review · Reviewer_Guwy · 2021-07-15

**Rating:** 5
**Confidence:** 3

**Summary:**

The paper presents a method to learn the dynamics of stable systems. The authors first exploit Neural Delay Differential Equations to model the partial observability and time delay of the system. They then employ Lyapunov-Razumikhin Function to ensure the stability of the learned dynamics. The authors evaluate their methods by presenting the sampled stable trajectories and loss plots of the learned model against other selected models.

**Limitations And Societal Impact:**

See main review.

**Main Review:**

[Strength] The paper has the following strength.
- The paper studies the modeling of the interesting type of time-delayed system and extends to partially observed stable systems.
- The proposed method is learning stable dynamics based on sampled trajectories in a data driven way.
- The method can be combined with policy learning to jointly learn a stabilizing policy and the dynamics of the closed loop system.

[Weakness] However, It has the following weakness about the paper which has stopped me from recommending the paper for publication in NeurIPS.
- The paper discusses little and conducts no comparison against the line of work from recurrent neural networks (RNNs) based effort on modeling system dynamics. Even though RNNs may lack the capability of modeling continuous dynamics, I think a more thorough discussion and empirical experiments here would increase the completeness of the work.
- The model introduces a few parameters to tune: K, samples of historical observations, r, the length of history to consider, and alpha, q for the exponential stability of the learned system. The parameters are highly tailored with the system dynamics to learn. That is, you would not be able to learn a good model for a (internally) higher dimensional partially observed systems with a small K (this is mentioned by authors as there would not be a one-to-one mapping from internal states to the coordinates). I am expecting a discussion on choosing the parameters and a demonstration/analysis of performance of the method against these parameters.
- The method bases on optimizing a loss function where a loss term is introduced to encourage stability. This only indirectly ensures stability. I am concerned that unseen observations could lead to non-stabilizing trajectories. If any theoretical guarantee could be established on this, it would increase the theoretical contribution of the work.
- The experiments has only compared the method against ANODE where ANODE's failure is shown through non-matching phase plots, but the success of the proposed method is only shown through lower loss. This alone is insufficient in showing a better performance of the proposed method as the worse performing ANODE with learned IC achieves lower loss than ANODE with true IC. More experiments could help make the point clearer.
- The method ports the established method of Neural Delay Differential Equations to the dynamics learning problem. The theoretical analysis of the method bases on established results clearly given in other papers -- The only theorem in this paper is given by (Efimov and Aleksandrov, 2020). The experiments have not shown any special benefits of the method against existing methods. Therefore, I am concerned that the novelty of the work is not up to the standard of NeurIPS.

_________________________________
Post rebuttal:
I have read the responses from the authors and appreciate the author's responses to my questions. However, I would like to stick with my original reviews and scoring.

**Time Spent Reviewing:**

8

---

> ### Author Response · Authors · 2021-08-10
> **Response to Reviewer Guwy**
>
> Thanks a lot for your feedback, valuable comments and helpful suggestions. In the following we would like to respond in detail to your comments. We addressed some of your concerns in our general comment to all reviewers, but would like to discuss some specific points you raised in the following.
>
> ## Our comments
> - __(C)__ *"The paper discusses little and conducts no comparison against the line of work from recurrent neural networks (RNNs) based effort on modeling system dynamics. Even though RNNs may lack the capability of modeling continuous dynamics, I think a more thorough discussion and empirical experiments here would increase the completeness of the work."* __(A)__ In lines 92-99 we quickly discuss how our NDDE approach is inspired by state augmentation in discrete-time models such as RNNs. Albeit discrete-time systems are both relevant and interesting, the focus of our work lies on pushing the current boundaries for continuous-time systems, motivating the choice of our comparisons.
>
> - __(C)__ *"The model introduces a few parameters to tune: K, samples of historical observations, r, the length of history to consider, and alpha, q for the exponential stability of the learned system. The parameters are highly tailored with the system dynamics to learn. That is, you would not be able to learn a good model for a (internally) higher dimensional partially observed systems with a small K (this is mentioned by authors as there would not be a one-to-one mapping from internal states to the coordinates). I am expecting a discussion on choosing the parameters and a demonstration/analysis of performance of the method against these parameters."* __(A)__ Thanks a lot for pointing out this ambiguity. As mentioned in our general comment we would like to add a section on this. Due to the following reasons the above parameters can be chosen without in-depth knowledge about the system at hand:
>     - As for the number of delays $K$, it is correct that we will not be able to learn a good model with too few delays. However, our experiments have shown that a high value of $K$ eases training and is not hurting generalization performance. Throughout the paper (and also the experiments in Appendix C.3) we thus use a relatively large number of delays $K=10$.
>     - Our choice for the magnitude of the largest delay $r$ is indeed differing in our experiments. However, note that Theorem 2([1]) in Appendix B is (besides the periodicity condition) agnostic to the choice of the delay parameter $\tau$. Nonetheless, in practice various heuristics such as *Average Mutual Information* or *False Nearest Neighbours* exist (for an overview see [2]).
>     - It is apparent from line 166-167 in Theorem 1([3]) that the parameter $q$ is directly related to the decay rate required by the Lyapunov-Razumikhin Theorem. This is underlined by our comments in line 228-230 and accordingly we are using $\alpha \approx \log(q)/r$ in our experiments.
>
> - __(C)__ *"The method bases on optimizing a loss function where a loss term is introduced to encourage stability. This only indirectly ensures stability. I am concerned that unseen observations could lead to non-stabilizing trajectories. If any theoretical guarantee could be established on this, it would increase the theoretical contribution of the work."* __(A)__ It is true that we only experimentally showcase the generalization performance of our approach (see Figures 3(b), 3(d), 4(a), 7, and 8(d)) and provide no theoretical discussion of generalization to unseen trajectories. To amend this, we will add a new theorem after line 241 in the paper, which establishes exponential decay based on continuous dependence of solutions (as outlined in detail in the comment to all reviewers).
>
> - __(C)__ *"The experiments has only compared the method against ANODE where ANODE's failure is shown through non-matching phase plots, but the success of the proposed method is only shown through lower loss. This alone is insufficient in showing a better performance of the proposed method as the worse performing ANODE with learned IC achieves lower loss than ANODE with true IC. More experiments could help make the point clearer."* __(A)__ This is a typo in the legend of Figure 2(d) (see comment to all reviewers). Thanks a lot for pointing this out! Thus, the Figure demonstrates the point that learning initial conditions is one of the main bottlenecks of ANODE. Does this sufficiently address your concern or did we miss something?
>
> - __(C)__ *"The method ports the established method of Neural Delay Differential Equations to the dynamics learning problem. The theoretical analysis of the method bases on established results clearly given in other papers -- The only theorem in this paper is given by [3]  . The experiments have not shown any special benefits of the method against existing methods. Therefore, I am concerned that the novelty of the work is not up to the standard of NeurIPS."* __(A)__ We added a discussion about novelty into the comment to all reviewers. While the approaches you mention certainly inspire our work, there are several technical subtleties when porting to continuous time, which require significant innovations.
>
> [1] Floris Takens, "Detecting strange attractors in turbulence"
>
> [2] Sebastian Wallot, Dan Mønster, "Calculation of Average Mutual Information (AMI) and False-Nearest Neighbors (FNN) for the Estimation of Embedding Parameters of Multidimensional Time Series in Matlab"
>
> [3] Denis Efimov, Alexander Aleksandrov "On estimation of rates of convergence in Lyapunov-Razumikhin approach"

---

> > ### Author Response · Authors · 2021-08-23
> > **short follow-up**
> >
> > We would like to thank reviewer Guwy again for their helpful comments and feedback. Please let us know if the above response was sufficient and in case there are no further questions, we would appreciate if you could update your score.

---

### Author Response · Authors · 2021-08-10
**General response to all reviewers**

Thank you for the valuable reviews, pointing out that the paper “studies the modeling of the interesting type of time-delayed system and extends to partially observed stable systems” (Guwy), “has the potential to apply to safety-critical systems” (Xxju) and “is the novel application of Lyapunov-Razumikhin stability to Neural DDEs” (2gyi). In the following we propose changes and answer to questions of general interest.

## Convergence guarantees
As pointed out by reviewers Guwy and 2gyi stability on unseen trajectories is currently only verified experimentally. To address this question, we will add the following Theorem establishing deterministic stability guarantees to line 241 in the paper (and its proof to the Appendix, see https://ibb.co/jfbZxkz):

### Theorem
Assume the time-delay system $\dot{x}=f(x\_t)$ has $L_f$-Lipschitz dynamics and that the LRF loss (16) is zero along trajectories starting in $\mathcal{S}\_{\text{train}}\subset\mathcal{C}\_r$ over a time horizon $[0,t_f]$. Then it is $(\gamma, M)$-exponentially decaying on $\mathcal{S}\_{\text{train}}$ over $[0,t\_f]$. Moreover, if for another set of initial histories $\mathcal{S}\supset \mathcal{S}\_{\text{train}}$, the training set $\mathcal{S}\_{\text{train}}$ is a $\delta$-covering (in the $||\cdot||\_r$-norm) of $\mathcal{S}$ with $\delta = e^{-(L\_f+\gamma)t\_f}\varepsilon$, then the time delay system is $(\gamma, \tilde{M})$-exponentially decaying on $\mathcal{S}\setminus B_\varepsilon(0)$ over the time horizon $[0,t\_f]$ and with $\tilde{M} = 2M+1$. Here, $B_\varepsilon(0) = \lbrace \psi\in\mathcal{C}\_r : ||\psi||\_r \leq \varepsilon \rbrace$ denotes the $\varepsilon$-Ball around the origin.

## New experiment
As suggested by reviewer 2gyi we add the stabilization of a delayed cartpole as a more complex experiment for stabilization with delayed feedback control. In contrast to the two-dimensional inverted pendulum this is a four-dimensional non-linear system (see [1] for the equations). We choose a delay of $\tau=0.05$ for the control force acting on the cart. As illustrated in the plot of the resulting test predictions (https://ibb.co/W5D8V61) the LQR is unstable, while the policies learned by our LRF loss remain stable.


## Novelty
While we rely on previously established theory, we believe that our work is by no means a trivial combination of previous results. More specifically it is novel in the following ways:

- While NDDEs have been used by [2] in a classification setting and by [3] as closure models for PDEs, our purpose and motivation are different. **We are the first to propose NDDEs to learn non-Markovian dynamical systems** such as they are occurring in partially observed or delayed dynamical systems. We discuss that our NDDE approach is inspired by state augmentation with memory states in discrete-time models. Moreover, in Appendix B we directly relate NDDEs to delay embeddings which, together with the universal approximation property of neural networks, results in sufficient conditions to represent partially observed periodic or chaotic dynamical systems.

-  Going from discrete-time delayed models to continuous NDDEs is non-trivial, since the system's state is becoming infinite-dimensional. This comes with theoretical as well as practical complications. Besides a more complicated stability analysis, we need an interpolation of the initial history and we therefore propose to **combine NDDEs with Gaussian process regression to learn a representation of the initial history**.

- Our experiments in Section 4 (lines 268-279) and Appendix C.3 (lines 587-600) **showcase that learning ANODEs with unknown augmented initial conditions is problematic** and leads to weaker training performance. In contrast, for our NDDE approach, the initialization is conveniently provided by the Gaussian Process mean function. Furthermore, the NDDE outperforms both the ANODE models in terms of training loss and number of training steps.

- Similarly as it has been shown in [4] for NODEs, we show that also NDDEs may become unstable for new initial histories. In order to prevent this we propose a regularization term based on Lyapunov-Razumikhin functions, where we propose to **discretize the Razumikhin condition** of Theorem 1([Efimov and Aleksandrov, 2020)]. This discretization is needed due to the infinite-dimensional state space. To the best of our knowledge this is the **first neural network based approach to stability analysis of time-delay systems** and is therefore quite significantly different from other neural Lyapunov papers that deal with the non-delayed case only.

- As appreciated by reviewer 2gyi, we also carefully show in Proposition 2 that our **discretization of the Razumikhin condition is not introducing too much conservatism.**

- In contrast to the non-delayed case, sampling in the infinite-dimensional state space $\mathcal{C}_r$ of NDDEs is non-trivial. As described in lines 242-256 we thus propose to **sample new initial histories in the finite-dimensional subspace of possible initial histories represented by a (bounded, Lipschitz) GP-mean function.**

- We showcase that our approach can **stabilize the learned model along unseen trajectories** and demonstrate that it can also be used to **learn stabilizing policies** in delayed feedback control.

## Other minor changes
- There has been a typo in the legend of Figure 2(d). We exchanged "learned IC" with "true IC". The loss of ANODEs with true initial conditions is lower than in the learned case (compare to Table 1 in Appendix C.2).

- As suggested by reviewer Guwy we would like to add a section about the choice of hyperparameters to Appendix C. The goal is to discuss our choice of parameters such as $K$ or $r$ and to make clear that those can be chosen without in-depth system knowledge. We will mention that a large number delays $K$ eases training and did not hurt generalization (we therefore fixed $K=10$) and that for the choice of $r$ well-known criteria from delay embeddings such as *Average Mutual Information* or *False Nearest Neighbours* may be used (for an overview see [5]).

[1] Stimac, Andrew K, "Standup and stabilization of the inverted pendulum"

[2] Qunxi Zhu, Yao Guo, Wei Lin, "Neural Delay Differential Equations"

[3] Abhinav Gupta, Pierre F. J. Lermusiaux, "Neural Closure Models for Dynamical Systems"

[4] Gaurav Manek, J. Zico Kolter, "Learning Stable Deep Dynamics Models"

[5] Sebastian Wallot, Dan Mønster, "Calculation of Average Mutual Information (AMI) and False-Nearest Neighbors (FNN) for the Estimation of Embedding Parameters of Multidimensional Time Series in Matlab"

---

### Decision · Program_Chairs · 2021-09-27

**Decision:**

Accept (Poster)

**Comment:**

Thank you for your submission to NeurIPS.  Although there is still some disagreement between reviewers after the rebuttal period, overall the proposed paper seems to be a solid contribution to the emerging field of modeling continuous-time dynamical systems with delays and/or partial observability.  The work is perhaps a bit incremental in that it is largely a combination of past approaches, including past work in Neural DDEs and past work on ensuring the stability of dynamical systems specified by Lyapunov functions.  But it also combines these past methods in new and rather interesting ways, and I believe is worthy of highlighting at NeurIPS.  Thus, while the paper is somewhat borderline, I am recommending it be accepted, with the strong comment that the authors address the reviewer concerns that _can_ be addressed (i.e., not just the general concerns about significance) as they did in the rebuttal.